# Growth and site-specific organization of micron-scale biomolecular devices on living mammalian cells

Sisi Jia[1], Siew Cheng Phua[2,6], Yuta Nihongaki [2], Yizeng Li[3,4], Michael Pacella[1], Yi Li[1], Abdul M. Mohammed[1], Sean Sun[3], Takanari Inoue [2] & Rebecca Schulman [1,5✉]

Mesoscale molecular assemblies on the cell surface, such as cilia and filopodia, integrate information, control transport and amplify signals. Designer cell-surface assemblies could control these cellular functions. Such assemblies could be constructed from synthetic components ex vivo, making it possible to form such structures using modern nanoscale self-assembly and fabrication techniques, and then oriented on the cell surface. Here we integrate synthetic devices, micron-scale DNA nanotubes, with mammalian cells by anchoring them by their ends to specific cell surface receptors. These filaments can measure shear stresses between 0-2 dyn/cm$^2$, a regime important for cell signaling. Nanotubes can also grow while anchored to cells, thus acting as dynamic cell components. This approach to cell surface engineering, in which synthetic biomolecular assemblies are organized with existing cellular architecture, could make it possible to build new types of sensors, machines and scaffolds that can interface with, control and measure properties of cells.

[1] Department of Chemical and Biomolecular Engineering, Johns Hopkins University, Baltimore, MD 21218, USA. [2] Department of Cell Biology, Johns Hopkins University School of Medicine, Baltimore, MD 21205, USA. [3] Department of Mechanical Engineering, Johns Hopkins University, Baltimore, MD 21218, USA. [4] Department of Mechanical Engineering, Kennesaw State University, Marietta, GA 30060, USA. [5] Department of Computer Science, Johns Hopkins University, Baltimore, MD 21218, USA. [6] Present address: Institute of Molecular and Cell Biology, Agency for Science, Technology and Research (A*STAR), Singapore 138667, Singapore. ✉email: rschulm3@jhu.edu

Micron-scale molecular assemblies including membrane-bound[1] and membrane-less[2] organelles, cilia[3], cytoskeletal networks[4], or the glycocalyx[5] spatially organize living cells, create specialized reaction environments, serve as transport conduits, and amplify chemical and mechanical signals in ways individual molecules cannot. Assembling synthetic micron-scale cell structures and controlling their dynamics are key goals of synthetic biology and nanotechnology[6] because these abilities could make it possible to construct, for example, new cellular reaction chambers, sensors, and information and material conduits.

A key challenge in this pursuit is that the formation and evolution of the cell's architecture are primarily kinetically driven[4]. The time-dependent concentrations of the assembling species must be controlled to build dynamic structures that integrate functionally into a cell's constantly evolving architecture.

A cell's architecture extends from its interior to its surface. Organizing and directing molecules on the cell surface is important for controlling cell phenotype and cell–cell interactions[7–10], drug and gene delivery, and building biotic–abiotic interfaces[11]. This can be achieved, for example, by attaching nanoparticles, small molecules[12], and nanowire cell–electronic interfaces[13] to the cell surface. While controlling interactions between cell receptors and nanostructures has been studied in the context of therapeutic modulation of receptor activity[14,15] and for directing import of therapeutics[16–19], less is known about creating and organizing microstructures that programmatically modify and extend cell surface architecture[7,20–22].

Micron-scale filaments are ubiquitous cell motifs that serve as sensors (antennae)[23], mechanical supports, agents for generating motion[24] and as substrates along which molecular motors can transport cargo. Filaments must grow and be anchored in prescribed orientations to execute these functions and actively grow and reorganize to maintain their structure and respond to stimuli. Here we organize micron-scale filaments that can act as functional cellular elements on specific cell surface receptors (Fig. 1a). We then grow these anchored filaments, demonstrating their capacity for dynamic reorganization. We also demonstrate how the cell-anchored filaments are sensitive flow rate meters whose dynamic range encompasses physiologically relevant rates of blood or ion channel-activated[25,26] flow. We thus show how micron-scale structures can be attached to a cell at specific locations in specific orientations in such a way as to extend the functional mesoscale architecture of the cell.

We used DNA tile nanotubes (Fig. 1b), which are semiflexible filaments with persistence length $8.7 \pm 0.5\ \mu m$[27] (on the order that of actin[28]) that polymerize via Watson–Crick hybridization[29,30]. These DNA nanotubes can grow from DNA origami templates, termed seeds (Fig. 1c)[27], and can reach $100\ \mu m$ in length[29–31]. Nanotube seeds serve as sites where nanotubes are anchored by their ends to specific substrates. DNA nanotube growth kinetics[30,32–35], hierarchical assembly pathways[36] and diffusion rates[27] have also been extensively characterized, allowing kinetic control over their growth and interactions with cells. Nanotubes can be functionalized with polymers, gold nanoparticles[37], proteins[38], and peptides[39], and thus could be templates for constructing diverse functional devices.

We sought an approach for anchoring DNA nanotube ends to specific receptors on living cells that could be easily tailored to target different receptor types and hypothesized that this anchoring could be performed by attaching receptors to nanotube seeds. The design of such an anchoring process presents key challenges. First, a nanometer-scale anchor point on a filament's end must bind specifically to the chosen receptor[40], and the filament's much larger remaining surface must not interact with the cell. Second, because nanotubes can be detached from or imported into an active cell, the attachment of nanotubes to cells must be achieved through kinetic control, where a nanotube's anchoring rate is higher than its rate of detachment or cell import[41]. Microparticles can be anchored to cells because their large surface areas lead to high net attachment rates[42,43]; molecules or complexes can be reliably anchored when they are supplied at high concentrations ($\gg 10\ nM$)[40]. Anchoring nanotubes requires interaction with a small area of nanotube surface, and because of DNA nanotubes' large size ($\sim 50\ MDa$), it is only practical to present them at concentrations $<100–200\ pM$. To overcome these challenges, we develop a method in which a DNA nanotube seed serves as an anchor and presents numerous binding sites that attach quickly and effectively irreversibly to the desired receptor. This approach yields efficient attachment to multiple receptors on multiple cell types with little nonspecific binding.

## Results

**Reliably anchoring nanotube seeds to cell receptors.** We first characterized and eliminated nonspecific interactions between DNA nanotube seeds and nanotubes and cells[44]. We measured the rate of DNA nanotube seed/cell interaction by adding Atto488-labeled DNA nanotube seeds (final concentrations 8–64 pM) to HeLa cells in culture (Supplementary Note S5). Confocal micrograph z-stacks showed that the average fluorescence intensity of seeds at the cells' midline increased linearly with seed concentration; the average fluorescence intensity for cells to which 64 pM seeds had been added corresponded to $107 \pm 17$ distinct attached seeds per cell (Supplementary Note S6, Supplementary Fig. S4a–c).

Poly(ethylene) glycol (PEG) coating can reduce nonspecific interactions between nanoparticles and cell membranes[45]. To test whether PEG coating might reduce nonspecific interaction between DNA nanostructures and cells, we hybridized 20 kDa PEG-15 nt DNA strand conjugates to seeds (Fig. 1c). Almost no PEG-coated seeds were visible on cells after 8–64 pM PEG-coated seeds were incubated with HeLa cells (Supplementary Fig. S4d–f).

We then conjugated 20 kDa PEG to nanotube monomers (Fig. 1d) and prepared seeded nanotubes by combining 415 nM PEG-conjugated monomers with 37 pM seeds and incubating them at 37 °C in TAE-$Mg^{2+}$ buffer for 3 days. $>40 \pm 4.8\%$ of the resulting filaments were $>3\ \mu m$ long (Supplementary Fig. S5), i.e. much longer than a micron. Neither PEG-coated nanotubes grown from PEG-coated or unmodified nanotube seeds attach to cells (Supplementary Note S8, Supplementary Fig. S6).

We first tried anchoring DNA nanotube seeds to cells using the SpyTag peptide and SpyCatcher protein, which form a covalent bond[46]. We hybridized six SpyTag peptide–DNA conjugates (Supplementary Note S9 and Supplementary Fig. S7 a, b) to each seed's barrel[27]. We then expressed a GFP–integrin–SpyCatcher fusion protein in HeLa cells via transfection (Supplementary Note S10). However, almost no nanotubes grew from SpyTag-modified seeds attached to cells (Supplementary Note S11 and Supplementary Fig. S7c, d), perhaps because of the low SpyTag–SpyCatcher reaction rate constant[46]: $1400 \pm 40\ M^{-1}\ s^{-1}$. Even assuming fusion receptor overexpression ($10^4$ per cell)[47], at 64 pM nanotubes on average just one nanotube would anchor to each cell per hour (Supplementary Note S12). Anchoring nanotubes requires a much faster binding reaction, so we next considered antibody–receptor interactions, as most protein interactions have forward rate constants of $10^5–10^6\ M^{-1}\ s^{-1}$.

We tried to anchor DNA nanotube seeds to epidermal growth factor receptors (EGFR) using EGFR–EGFR antibody binding. EGFR is a transmembrane receptor tyrosine kinase overexpressed at up to $10^6$ copies per HeLa cell[48]. As expected, fluorescently

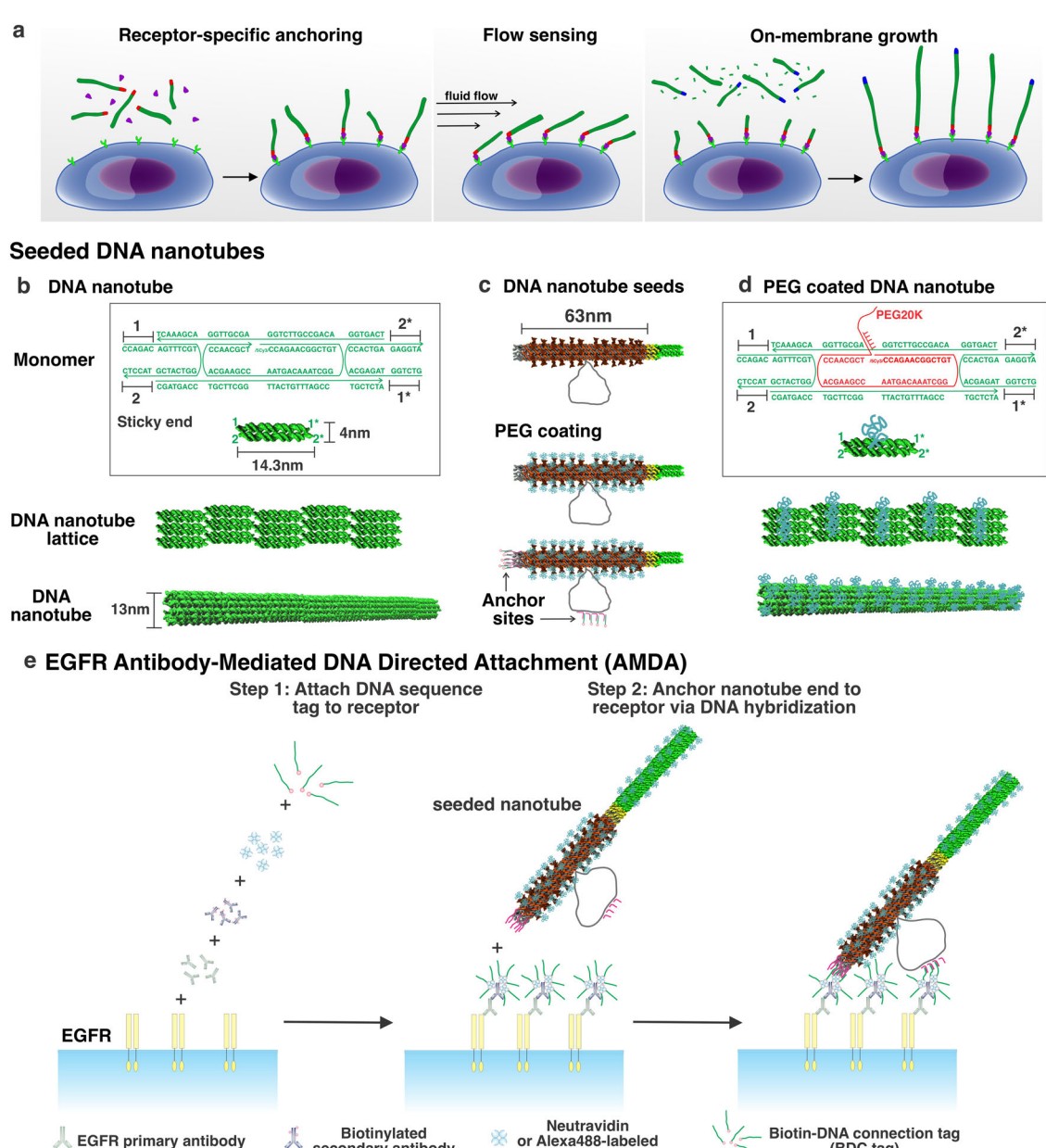

**Fig. 1 Anchoring synthetic filaments, DNA nanotubes, to specific cell surface receptors. a** DNA nanotubes anchored at specific locations on the cell surface could act as dynamic, functional elements of cells. **b** Micron-scale DNA nanotubes self-assemble from monomer complexes by hybridization of complementary single-strand DNA overhangs or sticky ends. Arrows indicate 3′ ends of DNA strands. X is complementary to X*. **c** Nanotube seeds are scaffolded DNA origami structures that template nanotube growth. Seeds can be coated with DNA–polyethylene glycol (PEG, molecular weight 20 kDa) conjugates (Supplementary Figs. S1 and S2). **d** PEG-coated DNA nanotube monomers and assembled nanotubes. **e** Schematic of EGFR antibody-mediated, DNA-directed attachment (AMDA) for anchoring seeded DNA nanotubes to cell surface receptors to origami seeds on nanotube ends. Primary antibodies, biotinylated secondary antibodies, streptavidin or neutravidin, and biotin–DNA connection tag (BDC tag) form complex to present a DNA sequence. This sequence hybridizes to the complementary DNA sequence on a DNA nanotube seed.

labeled secondary antibodies were attached to fixed and live HeLa cells only after EGFR primary antibodies were added (Supplementary Notes S13–S15, Supplementary Fig. S8). However, only a few DNA origami seeds with six secondary antibodies on their barrels[27] attached per cell (Supplementary Note S16, Supplementary Fig. S9). We realized that while receptor–antibody binding was likely fast enough for hundreds of seeds to attach, most antibodies (including the EGFR antibody[49]) have >nM affinities[40]. Provided at picomolar concentrations, DNA origami seeds would thus not remain attached on average.

We thus turned to DNA hybridization. Forward rate constants[50] of DNA hybridization is $10^5-10^6 \, M^{-1} \, s^{-1}$, and a

15-nt DNA strand binds to its complement with sub-picomolar affinity under physiological conditions (Supplementary Note S17). We designed a process in which a DNA sequence, termed the biotin–DNA connection (BDC) tag, is first attached to a receptor using antibodies, biotin, and neutravidin. Nanotube seeds then present the BDC tag complement (BDC′), which hybridizes to the tethered BDC (Fig. 1e). This approach can easily be generalized to attach different structures to different cell receptors: each of a set of antibodies could present a unique BDC sequence that would direct a particular unique nanostructure presenting that sequence's complement to bind. We termed this scheme "antibody-mediated, DNA-directed attachment" (AMDA).

To test AMDA, we first added >10 nM each of primary EGFR antibodies, biotinylated secondary antibodies, neutravidin, and the BDC tag in steps to live HeLa cells (Supplementary Table S10). We then added 16 or 64 pM nanotube seeds. These concentrations were the same as those used in earlier experiments, providing a clear basis for comparison with those experiments, and the use of two concentrations allowed us to observe whether the process might be seed concentration-dependent. About 2-fold more seeds attached to cells after adding either 16 or 64 pM seeds presenting six sequences complementary to BDC, BDC', at their barrels' ends than were attached after a control AMDA process where the BDC strand was not added (Supplementary Note S19, Supplementary Fig. S10).

We hypothesized that seeds' PEG coating might cover the BDC' sequences. We added 24 thymines to the BDC' presenting strands to increase the distance between the BDC' sequence and the PEG. We also replaced 30 fluorescently labeled DNA strands attached to a loop of DNA on the seed (Fig. 1c) with strands presenting the BDC' sequence. These changes dramatically increased the number of seeds attached to HeLa cells after AMDA without increasing nonspecific attachment (Fig. 2a, Supplementary Note S20). Cross-sectional images showed seeds on the cell membrane, consistent with receptor attachment (Fig. 2b). Elimination of any AMDA step almost completely eliminated seed attachment (Fig. 2c, Supplementary Fig. S11).

We next used AMDA to attach nanotube seeds to EGFR on suspended HEK293 cells (Supplementary Note S23). Nanotube seeds were present all over cells after AMDA, while little attachment was observed in controls (Fig. 2d). The fluorescence intensity over a background of HEK293 cells as determined by flow cytometry ($674 \pm 75$) was >5-fold higher after seeds were attached by AMDA than after a control process ($125 \pm 55$) (Fig. 2f, Supplementary Fig. S13 and Supplementary Note S24).

To verify that DNA seeds attached proximally to EGFR, we measured the colocalization of nanotube seeds with fluorescently labeled EGFR antibodies (Fig. 2g, h). $76 \pm 4\%$ ($N = 12$ cells) of seeds were colocalized with EGFR antibodies after AMDA; Stochastic attachment would result in only $20 \pm 2\%$ ($N = 12$ cells) colocalization (Fig. 2i, Supplementary Note S25).

Seeded DNA nanotubes attached reliably to HeLa cell membranes via AMDA but not in controls (Fig. 3a–c, Supplementary Note S26). Cells were 35-fold more fluorescent in the nanotube seed channel (Atto488) and 46-fold more fluorescent in the nanotube (Cy3) channel over background after AMDA vs. after a control protocol (Supplementary Note S27). Seeds did not move in time-lapse movies but attached nanotubes moved freely (Supplementary Movie S1), indicating that nanotubes were anchored to cells by seeds. Nanotubes could also be anchored to EGFR on HEK293 cells (Fig. 3d–f, Supplementary Movie S2, and Supplementary Note S28). They could also be anchored to integrin receptors on HeLa cells via AMDA, demonstrating AMDA's generality (Supplementary Note S30 and Supplementary Fig. S15).

We next asked how long nanotubes or nanotube seeds would persist on a cell's surface at 37 °C, as receptor turnover or seed endocytosis could lead to detachment or import. DNA origami seeds and seeded nanotubes were first attached to HeLa-GFP cells at 4 °C, where detachment rates were low. The cells were then returned to 37 °C where decreases in the number of seeds or nanotubes on the cell surface were measured using time-lapse confocal microscopy (Supplementary Notes S31 and S32). The fractions of nanotubes and nanotube seeds on the surface both decreased exponentially with time (Fig. 3g, h). The time constant for seeds was six times faster than for seeded nanotubes (Fig. 3h).

These measurements did not distinguish whether structures detached from or were imported into the cell[51]. EGFR-mediated endocytosis is a receptor-mediated clathrin-dependent pathway[52]

in which a membrane invagination is pinched off by the motor protein dynamin. Nanotube seeds (Fig. 1d, length: 65 nm) are small enough to conceivably be endocytosed with EGFR[53]. EGFR-mediated endocytosis takes on order 30 min[54], consistent with the rate of seeds leaving the cell surface. Nanotubes are too large to be endocytosed, but dynamin-controlled membrane closure might sever them. EGFR is a fast-turnover receptor[55] and HeLa cells are fast-growing cells so these persistence times are likely at the lower range across different receptors and cell lines.

**Nanotube shear stress sensors**. We next tested whether anchored nanotubes could measure cell surface shear stress. Shear can result from flow and is a key environmental signal in vivo. For example, the primary cilium[3] is involved in sensing flow in the kidney[56], and bends in response to flows[26,57,58], inducing signaling[59,60]. We asked whether, like primary cilia, nanotubes might bend in response to shear stress and whether the extent of this bending might indicate the magnitude of shear stress. Because nanotubes are microns in length, their bending angles could be precisely measured, suggesting that shear stress could be precisely determined as a result.

To assess this possibility, we developed a model of how a nanotube anchored to the surface of a rectangular chamber would respond to shear stress induced by laminar flow of velocity $U$ (Fig. 4a, Supplementary Note S34). A nanotube was modeled as a rigid rod anchored by a flexible linker. The chamber was much taller than a nanotube's length (see the "Methods" section), so the flow field around the nanotube should be essentially uniform (Fig. 4b). In simulations, the polar angle between the nanotube and z-axis was close to $\pi/2$ except at very small shear stresses, so we assumed this polar angle was $\pi/2$ under external flow (see Supplementary Note S34.2, Supplementary Fig. S22). The nanotube's response could therefore be reduced to an in-plane ($xy$ plane) rotation, i.e. the azimuth angle, $\phi$ between the nanotube and the flow's direction (Fig. 4b). In this case, the flow-induced viscous drag on the nanotube is $\mathbf{F} = (\alpha\mu U\ell, 0)$, where $\alpha$ is the coefficient of viscous drag on the nanotube, $\mu$ is the viscosity of the fluid in the chamber, and $\ell$ is the nanotube's length. The directional vector of the center of mass of the nanotube is $\mathbf{r} = (\ell/2 \cos\phi, \ell/2 \sin\phi)$. Thus, the torque on the nanotube is $\mathbf{M} = \mathbf{r} \times \mathbf{F} = -1/2\alpha\mu U\ell^2 \sin\phi$. We used this torque to calculate the dynamics of the nanotube, which are governed by $\gamma \, d\phi/dt = M + R$, where $\gamma$ is the nanotube's damping coefficient, $M = |\mathbf{M}|$, and $R$ is a random force from thermal fluctuations. $R$'s distribution is given by $P(R) \propto \exp[-R^2\Delta t /(2k_BT\gamma)]$, where $\Delta t$ is the time step used to numerically evolve the equation. For each time step, a random $R$ was drawn. The initial value of the azimuth angle, $\phi_0$, of each nanotube, was randomly drawn from the uniform distribution $[-\pi, \pi]$. We solved the probability distribution of $\phi$ by sampling $\phi$ for a large number of nanotubes for each a set of volumetric flow rates $Q = UHW$, where $H$ and $W$ are the chamber's height and width, respectively (Fig. 4c). We found that the distributions of azimuthal angles should vary for shear stresses between 0 and 1.5 dyn/cm$^2$, a range relevant for ion channel activation[25,26].

To measure the sensitivity and dynamic range of nanotube flow sensors, we anchored nanotube seeds to the bottom of a passivated glass microchannel[27] (Supplementary Notes S36 and S37) and measured their orientations under different flows using time-lapse spinning disk confocal microscopy (Fig. 4d, Supplementary Note S38). In the absence of flow, nanotubes explored all azimuthal angles and bent in the z-direction. Shear stress of only 0.05 dyn/cm$^2$ caused the nanotubes to remain in plane and align with the flow (Fig. 4d). To quantify the relationship between nanotube orientation and fluid shear stress on the glass, we

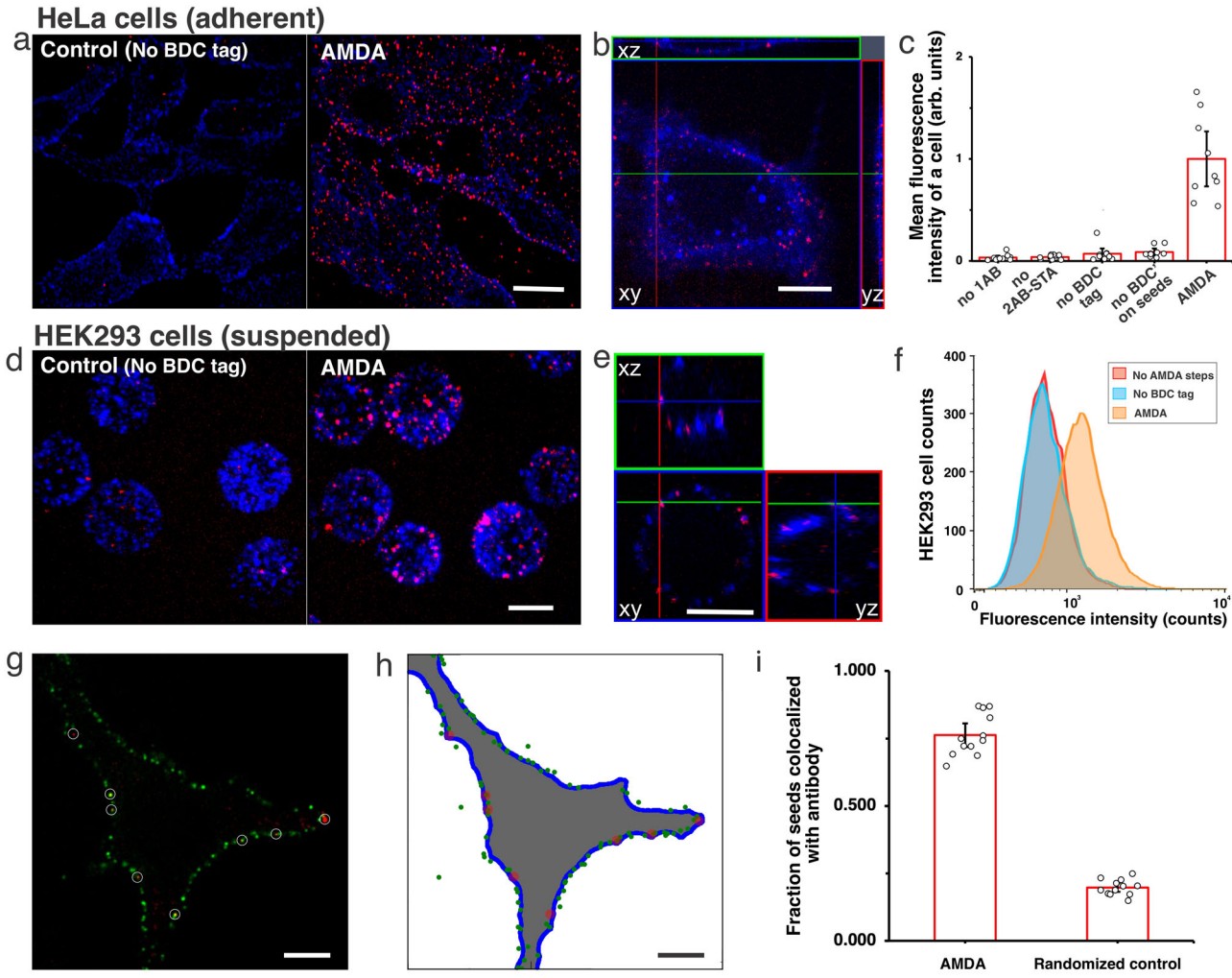

**Fig. 2 PEG-coated DNA nanotube seeds attach to EGFR receptors via AMDA. a**, **d** Three-dimensional projection images of HeLa (**a**) and HEK293 (**d**) cells after AMDA or AMDA with BDC tag addition omitted (No BDC tag) to attach seeds to EGFR (Supplementary Notes S20 and S23). Nanotube seeds were labeled with Atto488 (red) and secondary antibody–streptavidin conjugates with Alexa647 (blue). Scale bars: 20 μm. **b** Confocal micrograph cross-sections of HeLa cells stained with DiD dye (blue) (Supplementary Note S21) before seeds (red) were attached via AMDA. Scale bar: 20 μm. This experiment was repeated independently more than three times with similar results. **c** Average fluorescence intensities of nanotube seeds per HeLa cell after AMDA or after AMDA omitting different reagents with 32 pM seeded added (Supplementary Note S22, $N = 9$ random locations for each case were analyzed). For no 1AB: only the addition of EGFR primary antibody was omitted during the AMDA, no 2AB-STA: only the addition of secondary antibody and streptavidin conjugate (2AB-STA) was omitted, no BDC' tag on seeds: there were not BDC' tags modified on nanotube seeds. **e** Confocal micrograph cross-sections of HEK293 cells. Scale bar: 20 μm. This experiment was repeated independently more than three times with similar results. **f** HEK293 cell fluorescence in the channel used to label seeds after AMDA (orange), AMDA with BDC tag addition omitted (blue), and no AMDA (red), measured via flow cytometry (Supplementary Note S24). 48 pM seeds were added to cells. Average fluorescence intensities were 1423 ± 75 ($N = 9818$ cells), 883 ± 55 ($N = 9836$ cells), and 759 ± 6 ($N = 9867$ cells). **g**–**i** Co-localization of nanotube seeds and antibodies on the cell membrane (Supplementary Note S25). **g** Confocal micrograph cross-section of a HeLa cell after AMDA with secondary antibody labeled with Alexa647 (green) and nanotube seeds labeled with atto488 (32 pM) (red). Scale bar: 10 μm. **h** Computerized localization of antibodies (green) and seeds (red) of cells in (**g**). **i** Mean fractions of nanotube seeds colocalized with antibody after AMDA and in randomized controls ($N = 12$ cells were analyzed). Error bars here in (**c**), (**f**), and (**i**) are 95% confidence intervals.

measured the mean total angle of nanotube rotation over 30 frames taken every 5 s at different fluid shear stresses (Supplementary Note S41). A maximum time projection image of each nanotube was generated from these images indicating the nanotube's total range of angular motion, $\Phi$ (Fig. 4f). The mean $\Phi$ for different nanotubes experiencing given shear stress decreased with increasing shear stresses between 0.05 and 2 dyn/cm², which was consistent with our model's predictions (Fig. 4h).

Nanotubes attached to cells via AMDA (Supplementary Notes S39 and S40) also increasingly aligned with the flow as shear stress increased (Fig. 4e, g, and Supplementary Movie S3).

Because cells are not flat, a nanotube's location on a cell affects its bend direction and motion. The total angles of rotation of nanotubes on the tops of cells varied most in response to different flow rates. $\Phi$ for nanotubes on the tops of HeLa cells was close to both the values predicted by the model and the values measured on glass for all shear stresses, suggesting how anchored nanotubes can serve as "windsocks" on cells that indicate flow direction and the magnitude of shear stress the flow induces.

**Growing nanotubes on living cells**. A key advantage of using self-assembled biomolecular structures as cell surface microdevices

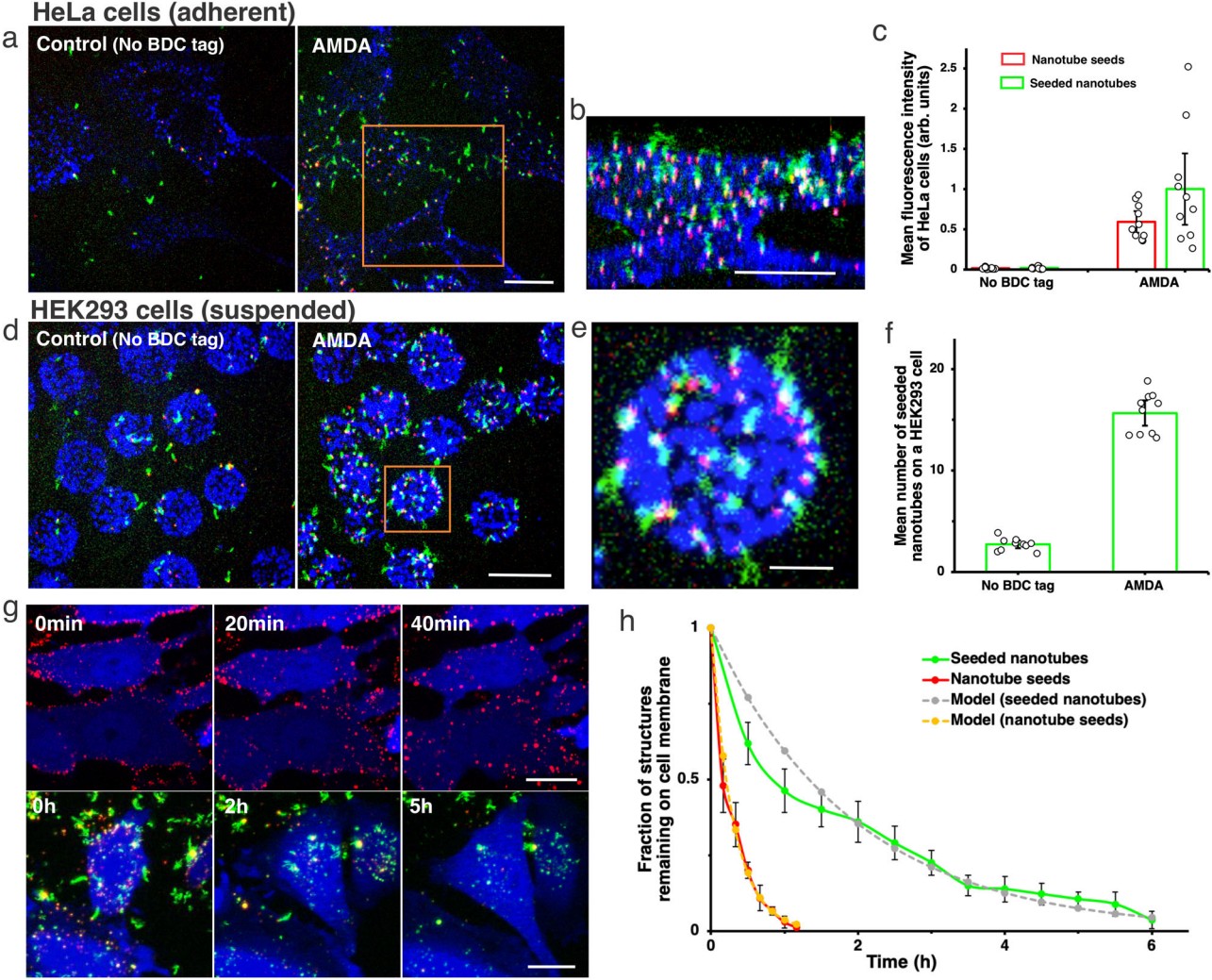

**Fig. 3 Anchoring DNA nanotubes to EGFR receptors using AMDA. a** Confocal micrographs of seeded nanotubes anchored to HeLa cells after EGFR AMDA and EGFR AMDA with BDC tag addition omitted (No BDC tag) (Supplementary Note S26). Seeds labeled with atto647 (red) and nanotubes with Cy3 (green), streptavidin with Alexa488 (blue). Scale bar: 20 μm. **b** Three-dimensional reconstruction of a HeLa cell with seeded nanotubes attached to EGFR. Scale bar: 20 μm. **c** The average fluorescence intensities of HeLa cells with seeded nanotubes or seeds attached via AMDA and AMDA with BDC tag addition omitted (Supplementary Note S27, $N = 10$ random locations for each case were analyzed). **d** Three-dimensional projection images of HEK293 cells with attached seeded nanotubes after AMDA and AMDA with BDC tag addition omitted (Supplementary Note S28). Scale bar: 20 μm. **e** Three-dimensional reconstruction of a HEK293 cell with attached seeded nanotubes. Scale bar: 5 μm. **f** Average numbers of seeded nanotubes on HEK293 cells after AMDA and after AMDA with BDC tag addition omitted (Supplementary Note S29, $N = 10$ random locations for each case were analyzed). The average numbers of attached seeds are shown in Fig. 2f. **g** Confocal micrographs of HeLa cells at different times after seed attachment using EGFR AMDA (upper panel) and maximum projection images of HeLa cells at different times after seeded nanotube attachment using EGFR AMDA (lower panel) (Supplementary Notes S31, S32). Scale bar: 20 μm. Three times of these experiments were repeated independently with similar results. **h** Fractions of DNA origami seeds or seeded nanotubes remaining on the cell surface at different times after AMDA ($N = 6$ cells for each case) (Supplementary Note S33). Fits are $y = ae^{-bt}$. For seeds, $a = 1.0 \pm 0.002$, $b = 3.3 \pm 0.50/h$, for seeded nanotubes $a = 1.0 \pm 0.002$, $b = 0.52 \pm 0.06/h$. Error bars here in (**c**), (**f**) and (**h**) are 95% confidence intervals.

is that they might dynamically grow or reorganize via biomolecular reactions. To explore the possibility of constructing such dynamic microdevices, we asked whether DNA nanotubes could grow while anchored to cell receptors.

Nanotubes can grow via monomer addition but at monomer concentrations where end-on growth is preferred over homogeneous nucleation, growth occurs at <0.2 μm/h[27,34,61]. Because nanotubes persist only a few hours on EGFR, we sought instead to extend nanotubes via the end-to-end joining of pre-assembled nanotubes[31,62]. While rapid end-to-end DNA nanotube joining has been observed in vitro[27], only $7 \pm 2\%$ ($N = 477$) of PEG-coated nanotubes underwent end-to-end joining within 4 h in cell buffer at physiological temperatures (Supplementary Note S42

and Supplementary Fig. S26). We hypothesized that end-to-end joining did not occur because the monomer detachment rate was very low, allowing rough facets or facets with defective monomers that cannot join to persist (Supplementary Fig. S28). To increase the monomer detachment rate, we shortened the monomers' binding sites from 6 to 4 nucleotides[63] to produce 4PEG nanotubes. $86 \pm 3\%$ ($N = 803$) of 4PEG nanotubes anchored to a glass surface grew via end-to-end joining within 3.5 h after 4PEG nanotubes (green) and 150 nM monomers that could serve as "glue" to fill in gaps between rough facets (red) were added (Supplementary Note S45 and Supplementary Fig. S29b). The average length of the joined nanotubes was $8.2 \pm 0.35$ μm ($N = 774$, Supplementary Fig. S31a). The 4PEG nanotubes

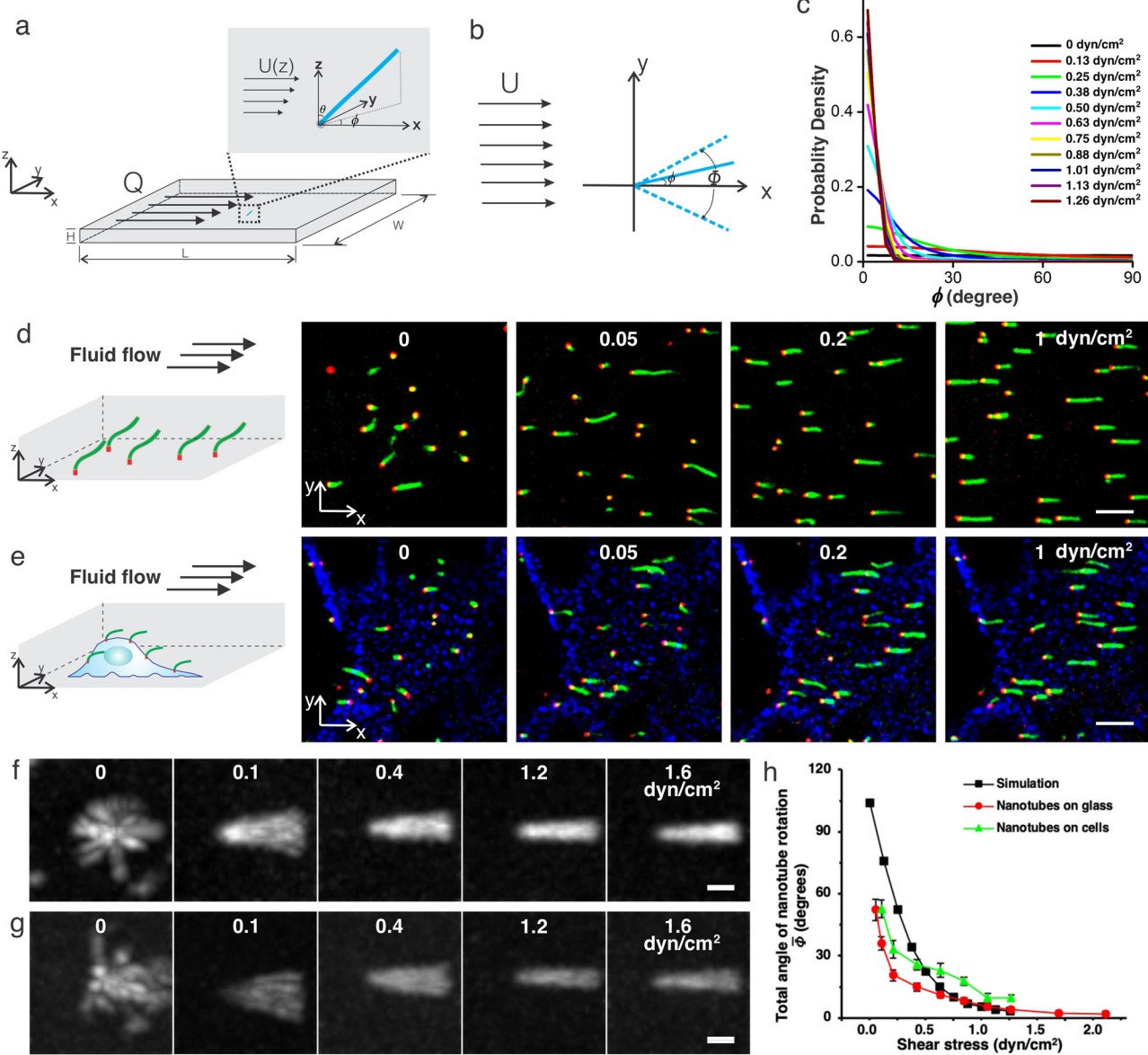

**Fig. 4 Anchored nanotubes indicate the magnitude of shear stress at the cell surface. a, b** Side (**a**) and top (**b**) views of nanotube deflection in a flow $Q$ in a rectangular channel. $\phi$ is the azimuthal angle between the plane of the nanotube and the $x$-axis. $\Phi$ is the total angle of nanotube rotation over a given time duration. **c** Predicted distribution of $\phi$ by a simple model of deflection (Supplementary Note S44.3). **d, e** Confocal micrographs of seeded nanotubes anchored on the glass surface of a rectangular flow chamber (see the "Methods" section) (**d**) and the top of HeLa cell membranes (**e**) in response to fluid shear stresses of 0, 0.05, 0.2, and 1 dyn/cm². Nanotubes were labeled with Cy3 (green), nanotube seeds with atto647 (red), and the cell membrane visualized with streptavidin-Alex488-conjugated EGFR antibodies (blue). Scale bars: 10 μm. More than three times of this experiment were repeated independently with similar results. **f, g** Maximum projection images of seeded nanotubes anchored on glass (**f**) and on HeLa cells (**g**) in response to fluid shear stresses 0, 0.1, 0.4, 1.2, and 1.6 dyn/cm². Scale bars: 2 μm. More than three times of this experiment were repeated independently with similar results. **h** Mean total angles of nanotubes as a function of fluid shear stress. $N = 15$ nanotubes for each shear stress on both glass and cells. Error bars here are 95% confidence intervals.

remained intact after end-to-end joining when the gentle fluid flow was applied to them (shear stress 0.32 dyn/cm²) for 35 min at 20 °C (Supplementary Fig. S29c).

60±9% ($N = 206$) of 4PEG nanotubes attached to EGFR on HeLa cells were extended to an average length of $6.9 \pm 0.62$ μm ($N = 189$, Supplementary Fig. S31b) using a similar protocol of end-to-end joining and monomer gluing (Fig. 5a–c, Supplementary Movie S4 and Supplementary Note S46). Since the flow used to image the joined nanotubes attached to cells sometimes severed the nanotubes (Supplementary Fig. S30), we repeated the joining process and then added methylcellulose to reduce

nanotube diffusion (Supplementary Note 48). This process revealed alternating green-red segments indicating nanotube gluing and joining (Fig. 5c–e, Supplementary Figure S33 and Supplementary Movie S5) as well as overlapping red and green segments indicating filament bundling, which can be induced by high viscosity medium[64].

## Discussion

While biomolecular filaments are structurally simple, they can assemble in myriad ways to create complex functional materials

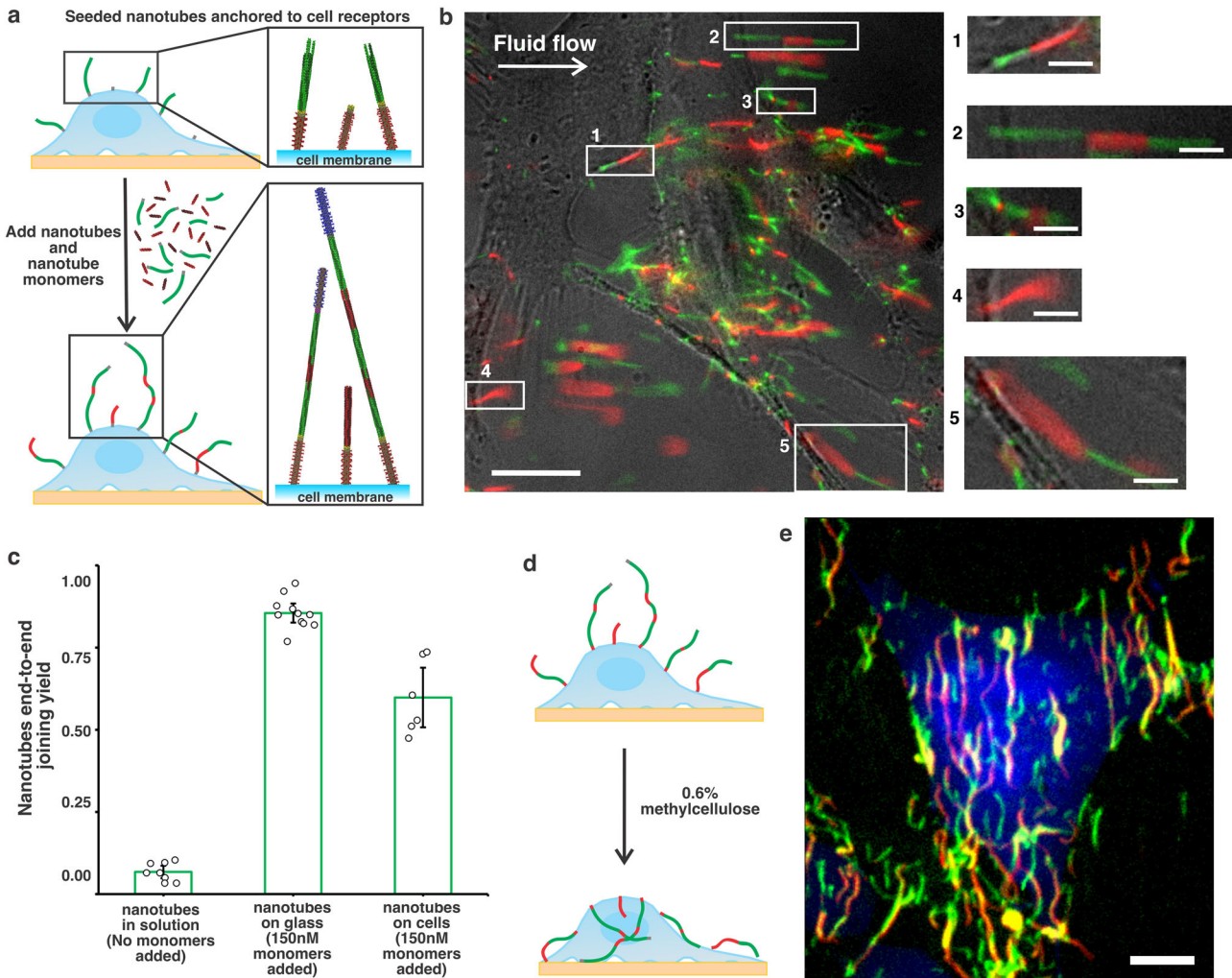

**Fig. 5 Nanotube growth on the surfaces of living cells. a** Schematic of nanotube growth via end-to-end joining and monomer addition. Seeds on anchored and capped nanotubes were unlabeled. Nanotubes were labeled with Cy3 (green), monomers with atto647 (red). **b** Two-color fluorescence micrograph of joined nanotubes anchored to live HeLa cells. Gentle fluid flow (shear stress 0.32 dyn/cm$^2$) was applied to stretch the nanotubes for better characterization (Supplementary Note S46). Scale bar: 20 μm. Zoom-in images of end-to-end joined nanotube structures on the cell surface. Scale bar: 5 μm. This experiment was repeated independently more than three times with similar results. **c** Nanotube end-to-end joining yield quantification: 6nt seeded nanotubes incubated with capped seeded nanotubes in solution at 37 °C for 3.5 h without additional monomers (Supplementary Note S42 and Supplementary Fig. S26, $N = 477$ nanotubes at $n = 8$ random locations was analyzed), 4PEG seeded nanotubes anchored on glass surface incubated with capped nanotubes and additional 150 nM monomers added at 20 °C for 3.5 h (Supplementary Note S45 and Supplementary Fig. S29, $N = 803$ nanotubes at $n = 11$ random locations was analyzed), 4PEG seeded nanotubes anchored on cell membrane incubated with capped nanotubes and additional 150 nM monomers at 20 °C for 4 h (Supplementary Note S46, $N = 206$ nanotubes at $n = 6$ random locations was analyzed). Error bars here are 95% confidence intervals. **d** and **e** Schematic (**d**) and 3D projection images (**e**) of joined nanotubes (green and red) on a live HeLa cell after 0.6% methylcellulose (IMDM) was added (Supplementary Note S48). Cells were transfected with GFP (blue) to reveal cell shape and extent. Scale bar: 10 μm. This experiment was repeated independently more than three times with similar results.

and devices, as exemplified by cytoskeletal structures. Here we site-specifically anchored synthetic DNA filaments to living cells by determining the required binding affinities, reaction rates and avidity for efficient attachment, and mitigating nonspecific interactions. These anchored DNA filaments act as cellular "windsocks" that can sensitively measure the magnitude of shear stress at the cell surface within a physiologically significant range. The resulting precise control over attachment-in combination with our understanding of DNA nanotube growth rates[30,34], hierarchical assembly[36] and reorganization[62]—might be used to build a range of synthetic, dynamic filament-based devices on cells, including antennae, motion-inducing devices or conduits that connect receptors on different cells. The ability to form nanotubes from many different types of monomers could also

make it possible to build more complex, heterogeneous devices.[37–39] Here we showed that nanotubes could remain attached to cells for multiple hours. Longer time scales might be achieved by attaching nanotubes to receptors with long turnover times.[65] The techniques developed here could also be combined with techniques that increase DNA nanostructures stability in biological environments, such as coating DNA nanostructures with cationic polymers[66], cationic block co-polymers[67], or lipid bilayers[68]. Nanotube lifetime in the presence of nucleases could also be increased by adding monomers to the cell medium that incorporate into nanotubes to repair damage[69]. We also demonstrate that, like the cells they are attached to, anchored DNA filaments can continue to grow as monomers or whole filaments can bind end-to-end to them. Growing filaments could

form between cells or anchor points. More generally, the understanding of how filament binding kinetics and thermodynamics, assembly timing, and component stoichiometry affect filament growth and organization might also enable the design of genetically encoded processes to direct the assembly of filaments synthesized by the cells themselves.

## Methods

**Reagents**. M13mp18 scaffold strand was purchased from Bayou Biolabs. All other DNA strands used in this study were synthesized by Integrated DNA Technologies, Inc. (IDT). Strands for the DNA nanotube tiles (Supplementary Note S49) and the adapter strands for the DNA nanotube seeds (Supplementary Note S50.3), Cy3-, ATTO647- and ATTO488-, biotin-labeled strands and amino-modified strands were HPLC purified. All other strands were simply desalted. Concentrations of DNA strands were determined either by measuring absorbance at 260 nm (using extinction coefficients supplied by IDT) or by relying on IDT to determine solution concentrations. N-hydroxysuccinimide (NHS) functionalized polyethylene glycol (molecular weight 20 kDa) (PEG-20K) was purchased from NANOCS (PG1-SVA-20K). Phosphate buffered saline (PBS) (28372) was purchased from ThermoFisher and prepared at 10× for further use. Gel loading dye blue (B7021S) was purchased from New England Biolabs and Sybr gold (S11494) was purchased from ThermoFisher. Centrifugal filters (UFC510096) for purifying seeds were purchased from MilliporeSigma. For cell culture, HeLa cell and HEK293 cell lines were both purchased from ATCC. DMEM medium (10-013-CV) was purchased from Corning Cellgro. FBS (26140079), 1% penicillin–streptomycin (15140122), 0.05% Trypsin–EDTA (25300054), and DPBS (14190144) were all purchased from ThermoFisher. For cell transfection, the Opti-MEM(1×) (31985062) was purchased from ThermoFisher and the X-tremeGENE (6365779001) from Sigma Aldrich. The azide-modified SpyTag peptide (Lot No. P3130-1) was synthesized by BioSynthesis. The size exclusion spin column Illustra Microspin G-25 was purchased from GE Healthcare. For AMDA, EGFR monoclonal antibody (H11) (MA5-13070), the Alexa fluor 647-conjugated secondary antibody (A-21236), the biotin-conjugated secondary antibody (31800), the streptavidin-Alexa Fluor 488 conjugate (S32354), neutravidin (31000), and DiD live-cell labeling solution (V-22887) were all purchased from ThermoFisher. Integrin β1 Antibody (K-20) (sc-18887) (for integrin AMDA) was purchased from Santa Cruz Biotechnology. Bovine serum albumin (BSA) (A3858) and MgSO4 were purchased from Sigma-Aldrich. The Streptavidin Conjugation Kit (ab102921), used for conjugating streptavidin with Alexa 647 labeled secondary antibody, was purchased from Abcam. Borosilicate glass Lab-Tek 8-well chambers (155411PK) were purchased from ThermoFisher. Glass-bottom dishes (μ-Dish 35 mm, high Grid-50 glass bottom) (81148), μ-slide VI 0.4 (80606), and glass-bottom μ-slide channel VI 0.5 (80607) were purchased from Ibidi. Biotin–PEG–silane (Biotin–PEG–SIL-3400−500 mg) was purchased from Layson Bio. Methylcellulose (HSC001) was purchased from R&D systems and Iscove's modified Dulbecco's medium (IMDM) (12440053), which was used to dilute the methylcellulose, was purchased from ThermoFisher.

**Synthesis of PEG–DNA strand conjugates**. 8 mg NHS-PEG20k was dissolved in 100 μL of a PBS buffer solution (pH 7.2) containing 50 μM amino-modified DNA strand. The mixture was agitated at room temperature (19–20 °C) overnight to allow the reaction to run to completion. Afterward, the solution containing the PEG20K–DNA conjugates was loaded into a 7% PAGE gel. The running buffer was TAE–Mg²⁺ (40 mM Tris-acetate, 1 mM EDTA to which 12.5 mM magnesium acetate was added) and loading buffer 1× blue gel loading dye. The gel runs at 150 V for 1 h. The desired band containing the PEG–DNA conjugate was cut out and the conjugate was extracted from the gel by soaking the gel in water for 2–4 days to let the conjugate diffuse out from the gel. The concentration of the PEG20K-DNA conjugate was determined by quantifying the amount of Cy3-labeled PEG-DNA conjugate using fluorescence intensity and using a known concentration to quantitate the unlabeled conjugate via PAGE gel (Supplementary Note S2).

**Assembly of PEG-coated DNA nanotube seeds**. The structure and sequences of DNA nanotube seeds used in this work are described in Supplementary Note S50 and Supplementary Tables S30–S35. To create PEG-coated DNA nanotube seeds, the staple sequences of a DNA origami seed structure[30] were modified to each present a DNA sequence that served as an attachment site for a PEG–DNA conjugate (sequence AAGCGTAGTCGGATCTC) (Supplementary Table S30). The resulting seeds were assembled, purified and their concentrations measured using protocols adapted from Agrawal et al.[34] (Supplementary Note S1 steps 2 and 3). To coat the resulting nanotube seeds with PEG, 18 μL of a solution containing 10 μM PEG-DNA conjugate and 1.8 μL 10× TAE-Mg²⁺ buffer were added to 100 μL of a TAE–Mg²⁺ solution containing 0.8 nM seeds and incubated on the bench for 30 min (see Supplementary Note S3).

**Growing PEG-coated seeded nanotubes**. The structures and sequences of DNA nanotube monomers with both 6- and 4-base sticky end binding sites are given in Supplementary Note S49 and Supplementary Tables S27 and S28. To grow PEG-

coated seeded nanotubes with six base sticky ends, the central SEs3 strand of the tile was conjugated with PEG as described in Supplementary Note S7. 19.7 μL of a TAE–Mg²⁺ solution containing 450 nM of each of the strands for the monomers were annealed from 90 to 37 °C as described in Supplementary Note S1 step 2. 2 μL of a solution containing 0.4 nM PEG-coated seeds was added after the monomer solution reached 37 °C. The mixture was kept at 37 °C for 3 days. To grow 4PEG nanotubes, the central REd3 and SEd3 strands of the two monomer types were each conjugated with PEG (Supplementary Fig. S36 and Supplementary Table S28). 19.7 μL of a TAE-Mg²⁺ solution containing 180 nM of each of the strands of the two 4PEG nanotube monomers was annealed from 90 to 20 °C as described in Agrawal et al.[34]. 2 μL of a TAE–Mg²⁺ solution containing either 0.4 nM PEG-coated anchored nanotube seeds (Supplementary Tables S20 and S21) or 0.4 nM PEG-coated capped nanotube seeds (Supplementary Table S22) as appropriate for experiments on nanotube joining were added after the solution reached 20 °C. The solution was then incubated at 20 °C for 3 days. Seeded nanotubes with PEG coating were imaged under an epi-fluorescence microscope (Olympus IX71) with a ×60/1.45 NA oil immersion objective lens using Andor SOLIS (Oxford Instruments) software.

**Cell culture**. HeLa cells (ATCC) and HEK 293 cells (ATCC) were grown in a DMEM medium containing 10% FBS and 1% penicillin–streptomycin. 5 mL of culture was grown in 25 cm² culture flasks at 37 °C in 5% CO2 and constant humidity. Cells were released from the flask surface using 0.05% Trypsin–EDTA and split every 2 days. The HeLa cells were cultured in media with: 3, 6, 9, and 12 mM MgSO4 overnight, after which time cell viability was confirmed by shape under a bright-field microscope (Supplementary Note S4).

**Characterizing the extent of nonspecific interactions between DNA nanotube seeds or seeded nanotubes and HeLa cells**. HeLa cells were seeded in borosilicate glass Lab-Tek 8-well chambers at a density of 40,000 cells per well in 250 μL medium (Supplementary Note S5 step 2). DNA nanotube seeds with and without PEG coating were diluted to make 8, 16, 32, and 64 pM solutions in a cold DMEM solution containing 1% BSA (w/v) and 12 mM MgSO4. The medium in each well chamber containing HeLa cell was exchanged for 250 μL of diluted seeds solution. The cells were then incubated in a 4 °C refrigerator for 30 min and subsequently washed with DMEM-12 mM MgSO4 buffer 3 times (Supplementary Note S5). Seeded DNA nanotubes grown from 37 pM seeds with PEG or 28 pM seeds without PEG in TAE–Mg²⁺ were diluted one-fold with cold 1% BSA(DMEM)–12 mM MgSO4. The medium in the wells containing HeLa cells was exchanged for diluted nanotube solution. The cells were then incubated in a 4 °C refrigerator for 2 h and subsequently washed with DMEM–12 mM MgSO4 3 times (medium in each well chamber Note S8). After both treatments, the cells were fixed using 4% paraformaldehyde before imaging.

**Transfection of HeLa cells with SpyCatcher-fusion transgenes**. The GFP-integrin-SpyCatcher plasmids were constructed by inserting the SpyCatcher DNA sequence into the GFP-integrin construct at the NotI restriction site. The backbone of the plasmid is a Clontech vector with kanamycin resistance. For plasmid sequences (see Supplementary Note S51). The plasmid was transformed and amplified in DH5alpha bacteria, and amplified using a Qiagen miniprep kit. HeLa cells for transfection were cultured and passaged as described above. HeLa cells were counted using a hemacytometer and diluted to 2.4 × 10⁵ cells per mL in DMEM. For each transfection process, 38 μL of Opti-MEM (1×) and 2 μL of a solution containing 1 mg/mL plasmid DNA were mixed well in a 1.5 mL Eppendorf tube via pipetting. 2 μL X-tremeGENE 9 solution was then added and the solution again mixed well via pipette, after which the mixture was incubated at room temperature for 30 min. 42 μL of the plasmid solution was then added to 500 μL of diluted cells in a 1.5 mL Eppendorf tube and mixed well by gently inverting the tube around 20 times. 250 μL of cell solution was then pipetted into a well of borosilicate glass Lab-Tek 8-well chamber. The cells were then incubated for 2 days before use (see Supplementary Note S10).

**Attachment of seeded nanotubes to cells using SpyCatcher-SpyTag attachment**. SpyTag-modified seeded DNA nanotubes were prepared by attaching an azide-modified SpyTag peptide to an amino-modified DNA oligonucleotide via a click reaction (Supplementary Note S9) and hybridizing this strand to the complementary sequence presented at the ends of nanotube seeds which were then spin-filtered to remove extra SpyTag (Supplementary Note S1 step 3). Nanotubes were then grown from seeds with and without the SpyTag modification and diluted 1.5-fold with DPBS–12 mM MgSO4 buffer. The medium in each well chamber containing HeLa cells expressed with GFP–integrin–SpyCatcher fusion protein was exchanged for 250 μL of diluted nanotube solution prepared as described for characterization of nonspecific nanotube–cell interactions but diluted by DPBS with 12 mM MgSO4 buffer and incubated at 37 °C/5% CO2 for 30 min (Supplementary Note S11).

**Attachment of nanotube seeds or seeded nanotubes to EGFR receptors on HeLa cells using AMDA**. PEG-coated nanotube seeds or seeded nanotubes were prepared as described as above and HeLa cell were seeded overnight in borosilicate

glass Lab-Tek 8-well chambers at a density of 40,000 cells per well with 250 μL medium (Supplementary Note S5 step 2). The next morning, the cells were first incubated in a 4 °C refrigerator for 10 min. DMEM buffer containing 1% BSA (w/v) was then added to the cells, which were then incubated for 5 min. To attach nanotube seeds, 250 μL of a solution containing (1) 2 μg/mL EGFR primary antibody, (2) 10 μg/mL Alexa 647-labeled secondary antibody–streptavidin conjugate, (3) 1 μM BDC tag and (4) BDC' tag-labeled nanotube seeds at concentration to achieve the stated concentration in cell solution were each added to the cells in the order listed. After each addition the cells were incubated in the refrigerator for 30 min then washed 3 times with cold DMEM (DMEM–12 mM MgSO$_4$ after seeds or nanotubes were added) to remove the reagent not attached to the cells. To attach seeded nanotubes, the Alexa 647-labeled secondary antibody solution was replaced by 250 μL solutions containing (2a) 500-fold diluted biotinylated secondary antibody followed by (2b) 3 μg/mL Alexa488-labeled streptavidin. After each addition cells were washed 3 times with cold DMEM buffer. To attach nanotubes to cells, the seed solution was replaced by a one-fold diluted solution of seeded nanotubes. This solution was incubated with cells for 2 h, pipetting gently every 30 min. After incubation cells were washed with DMEM–12 mM MgSO$_4$ buffer 3 times.

**Attachment of nanotube seeds or nanotubes to EGFR receptors on HEK293 cells using AMDA**. PEG-coated nanotube seeds and seeded nanotubes were prepared as described as above and HEK293 cells were trypsinized and suspended to a concentration of $10^6$–$10^8$ cells per mL. The cells were centrifuged at $300 \times g$ for 5 min, the supernatant was removed and the cells were resuspended in cold 1% BSA in DMEM. To attach nanotube seeds to suspended HEK293 cells, the cells were centrifuged and resuspended in a solution containing (1) 2 μg/mL EGFR primary antibody, (2) 10 μg/mL Alexa 647 2AB-STA, (3) 1 μM BDC tag, and (4) PEG-coated nanotube seeds in sequence. After each addition, the cells were incubated in a 4 °C refrigerator for 30 min and pipetted-mixed every 15 min, then washed by centrifuging and resuspending in 1 mL cold DMEM buffer to remove unattached reagent. After this sequence, the cells were resuspended in DMEM–12 mM MgSO$_4$ buffer. To attach seeded nanotubes, resuspension in Alexa 647-labeled secondary antibody solution was replaced by resuspension in (2a) 500-fold diluted biotinylated secondary antibody then (2b) 3 μg/mL Alexa488-labeled streptavidin. After the addition of the BDC tag and resuspension, the cells were resuspended in 1 mL cold DMEM buffer then a solution of seeded nanotube solution (1-fold diluted after preparation). The cells were incubated in a 4 °C refrigerator for 2 h during which time they were pipetted gently every 30 min. The cells were centrifuged a last time, then resuspended in DMEM–12 mM MgSO$_4$ buffer.

**Attachment of nanotubes to integrin receptors on HeLa cells using AMDA**. The steps for AMDA were followed above except that the EGFR primary antibody solution was replaced with a solution containing 4 μg/mL Integrin β1 antibody. After addition, cells were incubated at 4 °C for 1 h.

**Spinning disk confocal microscopy**. Cells with attached DNA nanotube seeds or seeded nanotubes were imaged using a Zeiss AxioObserver Yokogawa CSU-X1 spinning disk confocal microscope with a ×60 oil objective. Stack images were taken by ZEN2 (blue edition) from the bottom of the cell to the top of the cell with a stack depth of 0.27 μm (for HeLa cells) or 0.5 μm (for HEK293 cells) at 10–15 random locations.

**Quantification of the number of nanotube seeds and seeded nanotubes attached to each cell**. The average fluorescence intensity per cell was used to quantify the average number of seeds/seeded nanotube attached to a HeLa cell. A z-stack of images was collected at each imaging position, and the stack image from the height closest to the center of an average-sized cell was selected for analysis. This choice was made because this image largely excluded the structures attached to the glass rather than the cell while maintaining a sufficiently large cross-sectional area for each cell for analysis. The number/total length of nanotube seeds and seeded nanotubes present was measured by characterizing the total fluorescence intensity (Supplementary Notes S22 and S27). The seeds' intensity per cell and nanotube intensity per cell were then calculated by dividing these respective quantities by the number of cells in an image, which was also counted manually[70]. Flow cytometry was used to characterize the number of nanotube seeds on the HEK293 cells (Supplementary Note S24.2). The fluorescence intensity of ~10,000 cells were collected using a FACSCanto flow cytometer (BD Biosciences, USA) with BD FACS-Diva software v8.0 and analyzed by software FlowJo 10.4. The number of seeded nanotubes on HEK293 cells was determined by generating a 3-dimensional projection image from each of a z-stack of images collected at multiple random locations and manually counting the number of seeded nanotubes visible on each cell (Supplementary Note S29.2). The amount reported is the average of these counts.

**Quantification of the dwell time of seeds and seeded nanotubes on the cell membrane after AMDA**. PEG-coated DNA nanotube seeds or seeded nanotubes were anchored on HeLa–GFP cells using EGFR AMDA as described above. The cells were washed with cold (4 °C) DMEM medium–12 mM MgSO$_4$ then placed

into an incubator (37 °C, 5% CO$_2$ and constant humidity) on a Nikon A1 confocal microscope (Nikon, Tokyo, Japan) with a ×63 oil objective. Stacks of images at heights spanning the bottoms and tops of the cell in the field of view. Image stacks of cells with attached nanotube seeds were collected by NIS-Elements AR 5.02.01 (Nikon) every 10 min over 70 min with a stack height of 0.27 μm. Image stacks of cells with attached nanotubes were collected every 15 min over 20 h with a stack height of 1 μm. The number of nanotube seeds and seeded nanotubes on each tracked cell's membrane were counted manually. These numbers were normalized by the number of seeds or nanotubes on that cell at t = 0. Seeds were counted if they were visible at a cell's edge. The top and bottom images of a stack were omitted from quantification because it was not possible to determine whether seeds were beneath or above the cell rather than at the cell surface. The total number of seeded nanotubes on a cell's surface at each time point was manually counted using maximum projection images (Supplementary Note S33).

**Use of fluid flow to apply shear stress at a glass surface or HeLa cell membrane**. Shear stress was applied within flow cells (Ibidi, μ-slide VI 0.5 for glass-anchored nanotubes, height 0.54 mm, length 17 mm and width 3.8 mm and μ-slide VI 0.4 for cell-anchored nanotubes, 0.4 mm, length 17 mm and width 3.8 mm) and the shear stress corresponding to a given flow rate was calculated according to the methods provided by Ibidi (Supplementary Note S35 and Supplementary Table S17). Seeded nanotubes were anchored to the glass bottoms of flow cells using a method developed previously[27] (Supplementary Notes S36 and S37). A syringe pump (New Era, NE-1000) was used to induce controlled, unidirectional laminar flow. TAE-Mg$^{2+}$ buffer was used as flow perfusate (Supplementary Note S38). Seeded nanotubes were anchored to the HeLa cell membrane through EGFR AMDA performed in a flow cell (Supplementary Note S39). DMEM–12 mM MgSO$_4$ buffer was used as flow perfusate (Supplementary Note S40). Seeded nanotubes under fluid flow were imaged using a spinning disk confocal microscope with 5 s intervals for 30 cycles. The nanotubes were imaged in the xy-plane in which the largest number of seeds and the largest fraction of the nanotubes were in focus. The total angle of the nanotube rotation under fluid flow was measured by first cropping the area spanned by a single nanotube from each larger image. A Gaussian blur filter (radius:1.00) was applied in ImageJ for all the 30 cropped images to reduce the image background. A maximum time projection image was then generated from this cropped time-lapse movie and the total angle of the nanotube rotation under fluid flow was measured manually (Supplementary Note S41). The total angle of 15 nanotubes on the glass surface and >15 nanotubes on cell membrane was measured for each shear stress.

**Growth of seeded nanotubes anchored to the cell membrane**. 4PEG nanotubes and 4PEG capped and seeded nanotubes were prepared as described in Supplementary Note S43. Here both the capped and anchored nanotube seeds were unlabeled (Supplementary Tables S20 and S22). Anchored seeded nanotubes were attached to the HeLa cell membrane through EGFR AMDA within a μ-slide channel. 50 μL of a solution containing 900 nM inactive nanotube monomers (Supplementary Note S44 and Supplementary Table S29) labeled with atto647 (Supplementary Table S25) was annealed in TAE–Mg$^{2+}$ buffer from 90 to 20 °C. 0.27 μL of solution containing 100 μM of a strand to activate the inactive monomers (Supplementary Note S46) was added to this solution after which 25 μL of it was immediately mixed with 60 μL of a solution containing 37 pM of capped nanotubes and 65 μL of DMEM–12.5 mM MgSO$_4$ buffer containing 1% BSA. This mixture was immediately added to the HeLa cells to which seeded nanotubes had been attached. The sample was covered with foil and incubated on the lab bench (at about 19–21 °C) for 4 h. It was then washed with DMEM–12 mM MgSO$_4$ three times before imaging. A gentle fluid flow (0.18 mL/min) inducing a shear stress of 0.32 dyn/cm$^2$ was applied to stretch the nanotubes and allow visualization of their contours. An epi-fluorescence microscope with a ×60 oil objective was used to capture continuous 20 images of each location. The yield of nanotube joining on cell membrane was calculated by counting the total number of seeded nanotubes on the cell and the number of nanotubes on the cell that had visually joining. Cells in six images were quantified. 0.6% methylcellulose media (IMDM) with 12 mM MgSO$_4$ was added to HeLa GFP cells after the joining protocol (but not flow or imaging) was completed (Supplementary Note S48). Stacks of images at random locations were taken from the bottoms to tops of cells with a stack height of 0.27 μm using a spinning disk confocal microscope.

**Reporting summary**. Further information on research design is available in the Nature Research Reporting Summary linked to this article.

## Data availability:
The data for this study are available online at: https://doi.org/10.7281/T1/CUEL0W. Source data are provided with this paper.

## Code availability:
The simulation code and image analysis code for this study are available online at: https://doi.org/10.7281/T1/CUEL0W. Source data are provided with this paper.

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

## Acknowledgements

The authors thank Samuel Schaffter, Yi Li and Pepijn Moerman for helpful discussion, Kostas Konstantopoulos, Panagiotis Mistriotis, and Kaustav Bera for technical assistance and HeLa-GFP cell lines, Michael McCaffrey and Erin Prycell for imaging assistance. S.J., Yi Li and R.S. acknowledge support from DARPA BTO Award D16AP00147 (YFA) and NSF CMMI-1562661. This work was supported in part by the US National Institutes of Health (NIH) grant to T.I. (DK102910). S.C.P. was supported by the Agency for Science, Technology and Research (Singapore). Y.N. was supported by postdoctoral fellowships from Japan Society for the Promotion of Science and from the Uehara Memorial Foundation. Yizeng L. and S.S. acknowledge support from NIH R01GM134542.

## Author contributions

This study was conceived by S.J., R.S and T.I. S.J. designed the experiments, carried out the experiments, and analyzed and interpreted the data. S.C.P. and A.M.M. designed the SpyTag-SpyCatcher fusion binding approach for nanotube-cell receptor attachment. S.C.P. constructed the GFP-integrin-SpyCatcher and GFP-integrin plasmids DNA and helped with cell culture and cell transfection. Y.N. helped with the flow experiment setup. M.P. and Yi. L. helped with data analysis. Yizeng. L. and S.S. designed the flow model, performed the simulation and analyzed the simulation results. S.J. and R.S. wrote the manuscript with input and edits from all authors.

## Competing interests

The authors declare no competing interests.
