## [Peer Review File · Nature Communications]

Reviewers' Comments:

Reviewer #1:

Remarks to the Author:

This is an excellent manuscript, describing the attachment of DNA origami nanotube seeds to cellular receptors and growth of DNA tile nanotubes on live cells. Being able to create and grow DNA nanostructures on cells will allow their use to both manipulate and report on cellular activity. While other DNA nanostructures have been attached to cell surfaces, I have not seen the growth and assembly of structures on live cells. The work is of extremely high quality; troubleshooting steps are thoroughly explained and will be very helpful to the community. The authors should comment on the stability of the grown nanotubes, when they switched to 4-nucleotide binding sites. Importantly, the serum stability and degradation profiles will play a major role in the growth rates, the disappearance of signals and the integrity of the nanotubes; this should be examined and discussed. I recommend acceptance of this manuscript with these minor revisions; it was a pleasure to read.

Reviewer #2:

Remarks to the Author:

The manuscript describes an experimental approach enabling attachment of synthetic DNA nanotubes to live cells. In contrast to purely synthetic self-assembled systems, attachment of man-made nanostructures to living cells presents several unique challenges. The authors have overcome many of them by using two sets of antibodies and a customizable DNA-biotin handle, which, in principle, allows targeting multiple cell receptors with unique nanostructures. Through fluorescence microscopy experiments, the authors have shown that such nanostructures indeed selectively attach to target receptors at the cell surface and can serve as local probes of fluid flow. Finally, the authors show that DNA nanotube self-assembly can proceed starting from seed nanostructures attached to the surface of live cells, which is a considerable technological accomplishment. The manuscript is clearly written. The conclusions are supported by numerous supporting information figures. The scope and the level of new results is appropriate for publication in Nature Communication.

Reviewer #3:

Remarks to the Author:

The authors of this study present a novel approach for attaching mesoscale molecular assemblies onto living cell surfaces. The noteworthy results include: 1) designed DNA nanotubes for cell surface receptor specificity using antibody-mediated attachment, 2) cell-anchored nanotubes as shear sensors on the surface of living cells, 3) dynamic growth/extension and reorganization of the nanotubes on live cells. The authors demonstrated that this attachment approach works on HeLa as well as HEK293 cells. Overall, the study is a very intriguing to read. However, there are a number of places where it would benefit from clearer explanations to alleviate some ambiguity to the readers. Below are comments for the authors' consideration:

Comments on the abstract

1. A sentence following "The challenges in creating such devices" could be added: a specific current challenge in either creating cell receptor-specific nanotubes or using nanotubes as flow sensors, which would transition into results the papers addresses (that have already been written clearly)
2. It may be an overstatement that nanotubes are maintained "during active cellular function" if most do not remain on the cell surface after a few hours (Fig 3h).

Comments on the Introduction:

3. The first two paragraphs under the Results section in my opinion belong in the Introduction. There should be mention of DNA nanotubes, as well as emphasis on an important point missing in the introduction: that the nanotube seeds aid "attachment to multiple receptors on multiple cell

types with little nonspecific binding.”

4. Restructuring of the Introduction section and first two paragraphs of the Results section is needed. Suggestions to the authors include:

a. Although a reference is made to Figure 1c for what DNA nanotube seeds are, its purpose is not clear until the end of paragraph two: “DNA nanotube seed serves as an anchor and presents numerous binding sites that attach quickly and effectively irreversibly at the desired receptor.”

b. Mention of the cell architecture as “primarily kinetically driven” could be merged with discussion of “the nanotube’s anchoring rate.”

5. The introduction presents a list of brief examples could be replaced with justification of the experimental approaches. A suggestion for the author’s consideration, the sentence starting with “Organizing and directing molecules on the cell surface is important for controlling cell fate” could be removed. Instead, having a few detailed examples of the significance and current unknowns of “creating and organizing microstructures that programmatically modify and extend cell surface architecture” and “cell-anchored filaments are sensitive flow rate meters” would in my opinion strengthen the conclusions of the paper.

6. To the point above, there are reported examples of functionalizing living cell membranes, such as the work by Akbari et al, *Adv Mater*, 2017 where synthetic DNA scaffolds were anchored to the surfaces of adherent, suspension, and primary cell types on which numerous functions such as directed cell-cell contacts and fluorescence readouts could be programmed.

Comments on the Results (reliably anchoring nanotube seeds to cell receptors)

7. In the results section, the authors mention that the number of attached seeds per cell was 107 ± 17 seeds for 64 pM seeds. However, it is not very clear how they came to that estimation. Looking at the supplementary material, it looks like the authors were able to measure the average fluorophore intensity per cell. However, making a direct link between the intensity and the number of seeds can be misleading. A better estimate can be the average concentration of seeds per cells, since this is the standard for the titrations made.

8. In the EGFR-EGFR antibody binding scheme, is there any reason why the authors did not increase the concentration of the seeds to nM scale to show that the limiting factor was indeed the affinity?

9. Is there a reason for the 16 and 64 pM seed concentration for the AMDA scheme?

10. Is there a reason why the quantification of the nanotube attachment to HEK293 suspended cells was done using flow cytometry as opposed to direct quantification from confocal images? How do the confocal results compare to the flow cytometry?

11. On a similar note, what is the reason of using two different standards for reporting the anchoring of DNA nanotubes in figure 3 c, f (i.e. mean fluorescence intensity vs. seed count)

12. In measuring the stability of DNA nanotubes on cell surface, what is the effect of the cell suspension buffers on the stability of the nanotubes?

13. On a similar note, what is the effect of the anchoring of DNA nanotubes, and the later addition of methylcellulose, on cell mortality?

14. “ $>40 \pm 4.8\%$ of the resulting filaments were $>3 \mu\text{m}$ long.” An explanation of the significance of this exact length would be fitting.

15. Supp. Figure S5b: Is there a reason incubation couldn’t be extended to 3 days? It appears that approximately 40% of filaments grown in these conditions were also more than 3 μm but was left out of the main text.

(Comments on the Results (shear stress study):

16. What is the height of the flow chamber? In the main text, it was mentioned to be 400 μm height. However, in the caption of figure 4, it was mentioned to be 0.54mm in height.

17. On a similar note what are the dimensions for the flow chambers used for the cell experiments? What is the reason for the difference between the formulas 4.31 and 4.32 in the supplementary material?

18. Minor comment: was the flow channel surface's hydrophobicity/hydrophobicity taken in consideration while assuming a uniform flow?

19. On a similar note, for the cell surface measurements, were the disruptions in the flow profile due to the presence of cells taken in consideration while estimating the shear stress? It appears like the shear stress reported in this case was still the wall shear stress as opposed to the estimated shear stress on the cell surface.

20. How were the nanotubes on top of the cells selected? Since the HeLa cells are of irregular topography, what defines the top of the cell?

21. Looking at figure S22, it seems like the angles ϕ_{upper} and ϕ_{lower} were measured from different points on the y-axis. Assuming that the seed doesn't move over time, would not it be more accurate to measure both angles with respect to the same anchoring point? (i.e. the seed)

22. What was the criteria for selecting the nanotubes to be analyzed at each shear stress value? Did they have to have a similar X-Y projection length? What happens with the nanotubes that appear as dots at lower flow rates?

Comments on the Results (Growing nanotubes on living cells)

23. Reasoning for using 4PEG nanotubes stated in Supp. Note 44 should also be stated in main text. They provide "higher attachment and detachment rate than the 6nt seeded nanotubes monomers."

24. A histogram of average lengths of nanotubes after extension would be a good visual to include.

25. Are there applications for the random specificity of end-to-end joining of nanotubes (Figure 5b)?

26. Maybe this isn't possible in the presence of certain flow rates, but since growing nanotubes can extend to micron-lengths, have any nanotube loops been observed e.g. shown below?

Comments on the conclusions:

27. Data interpretation is somewhat limited. Discussion could include a summary of the main results, limitations of a low fraction of seeded nanotubes remaining on the cell surface after several hours and not being able to control the final length of growing nanotubes, and how AMDA compares to and leads other current antibody mediated DNA attachment devices, if any.

Comments on the figures and supplementary movies:

28. Figure 1: (A) The cell receptors are hard to see against the background. Changing the color might help making them more distinguishable. (B) The panel would benefit from additional annotation to show that radial and longitudinal polymerization of the nanotubes. (C) What are the dimensions for the seeds?

29. Figure 2: Are these results for the 64 pM seed concentration? If yes, please add that to the caption

30. Figure 4: (A) The nanotube deflection is barely visible. (D, E) The confocal images in panels d and e should have axes

31. Supplementary Figure S5: Looking at the sample epi-fluorescence of the seeds after incubation, is there a reason why the polymers look more coiled after 3 days of incubation as compared to just one day even though there is not a big shift in the length distribution? Were the same samples imaged after 1 and 3 days of incubation?

32. Time stamps should be added to Supp. Movies 3 and 4

Reviewer #1 (Remarks to the Author):

This is an excellent manuscript, describing the attachment of DNA origami nanotube seeds to cellular receptors and growth of DNA tile nanotubes on live cells. Being able to create and grow DNA nanostructures on cells will allow their use to both manipulate and report on cellular activity. While other DNA nanostructures have been attached to cell surfaces, I have not seen the growth and assembly of structures on live cells. The work is of extremely high quality; troubleshooting steps are thoroughly explained and will be very helpful to the community.

The authors should comment on the stability of the grown nanotubes, when they switched to 4-nucleotide binding sites. Importantly, the serum stability and degradation profiles will play a major role in the growth rates, the disappearance of signals and the integrity of the nanotubes; this should be examined and discussed. I recommend acceptance of this manuscript with these minor revisions; it was a pleasure to read.

Comments/remarks:

We appreciate the reviewer's kind remarks about this work, the fact that the detailed information in the paper will be helpful to the community at large and the interest of the work and its presentation.

We agree that the stability and degradation of the nanotubes is of clear importance. We now mention the stability of the nanotubes with 4-nucleotide binding sites (where presumably thermal melting would be a concern) during and after being combined with cells.

We also agree that the question of how the stability of the DNA nanostructures could be affected by the biological environment where these structures are used is also important to examine. The experiments in this work were performed in serum-free cell medium (1% BSA in DMEM medium with 12mM MgSO₄) and nanotubes were incubated with cells in fresh medium over relatively short periods (*i.e.* no more than a few hours). In these experiments, we did not observe any significant degradation of nanotubes. In other experiments, however, with serum-containing cell media, during experiments spanning longer periods where nucleases and other products may build up in culture or *in vivo*, nanotubes may undergo nuclease-mediated or other forms of degradation. Mention of these observations are now included with citations in the paper. We now describe in the conclusion how the methods we develop could also be combined with those designed to combat degradation of nanostructures in these environments over time. Notably, the experiments in which we add monomers to nanotubes to grow them suggests that one such method for increasing nanostructure lifetime in serum, the addition of monomers to a solution which can integrate into nanotubes and thereby repair them, might be used with essentially no modifications to our current process.

Reviewer #2 (Remarks to the Author):

The manuscript describes an experimental approach enabling attachment of synthetic DNA nanotubes to live cells. In contrast to purely synthetic self-assembled systems, attachment of man-made nanostructures to living cells presents several unique challenges. The authors have overcome many of them by using two sets of antibodies and a customizable DNA-biotin handle, which, in principle, allows targeting multiple cell receptors with unique nanostructures. Through

fluorescence microscopy experiments, the authors have shown that such nanostructures indeed selectively attach to target receptors at the cell surface and can serve as local probes of fluid flow. Finally, the authors shows that DNA nanotube self-assembly can proceed starting from seed nanostructures attached to the surface of live cells, which is a considerable technological accomplishment. The manuscript is clearly written. The conclusions are supported by numerous supporting information figures. The scope and the level of new results is appropriate for publication in Nature Communication.

We appreciate the reviewer's kind remarks about this work.

Reviewer #3 (Remarks to the Author):

The authors of this study present a novel approach for attaching mesoscale molecular assemblies onto living cell surfaces. The noteworthy results include: 1) designed DNA nanotubes for cell surface receptor specificity using antibody-mediated attachment, 2) cell-anchored nanotubes as shear sensors on the surface of living cells, 3) dynamic growth/extension and reorganization of the nanotubes on live cells. The authors demonstrated that this attachment approach works on HeLa as well as HEK293 cells. Overall, the study is a very intriguing to read. However, there are a number of places where it would benefit from clearer explanations to alleviate some ambiguity to the readers. Below are comments for the authors' consideration:

We appreciate the reviewer's positive thoughts of this work and the novel results presented.

Comments on the abstract

1. A sentence following "The challenges in creating such devices" could be added: a specific current challenge in either creating cell receptor-specific nanotubes or using nanotubes as flow sensors, which would transition into results the papers addresses (that have already been written clearly)

We appreciate the reviewer's thoughts about the abstract could be improved. We have clarified the abstract to improve the transition between the general goals and our results in line with these suggestions.

2. It may be an overstatement that nanotubes are maintained "during active cellular function" if most do not remain on the cell surface after a few hours (Fig 3h).

We appreciate the reviewer's perspective here and we have revised the language in the abstract to eliminate this specific claim.

Comments on the Introduction:

3. The first two paragraphs under the Results section in my opinion belong in the Introduction. There should be mention of DNA nanotubes, as well as emphasis on an important point missing in the introduction: that the nanotube seeds aid "attachment to multiple receptors on multiple cell types with little nonspecific binding."

This is a good suggestion – these paragraphs do not contain results but rather clarify the problem the paper seeks to solve. We have therefore moved the first two paragraphs of the results

section. Moving these paragraphs also adds the points that the reviewer suggests into the introduction.

4. Restructuring of the Introduction section and first two paragraphs of the Results section is needed. Suggestions to the authors include:

a. Although a reference is made to Figure 1c for what DNA nanotube seeds are, its purpose is not clear until the end of paragraph two: “DNA nanotube seed serves as an anchor and presents numerous binding sites that attach quickly and effectively irreversibly at the desired receptor.”

This is a good suggestion. We fully agree that not mentioning the function of the nanotube seeds in the context of the present work was an oversight on our parts. In the revised version of the manuscript, we have added this point when introducing the seeds and re-emphasize it in the following paragraph when we mention attachment of nanotubes to specific receptors by their seeds.

b. Mention of the cell architecture as “primarily kinetically driven” could be merged with discussion of “the nanotube’s anchoring rate.”

We agree that discussing the kinetic process and nature of attachment in the context of the nanotube anchoring rate is a good one. We have now added this to the introduction in the revised version.

5. The introduction presents a list of brief examples could be replaced with justification of the experimental approaches. A suggestion for the author’s consideration, the sentence starting with “Organizing and directing molecules on the cell surface is important for controlling cell fate” could be removed. Instead, having a few detailed examples of the significance and current unknowns of “creating and organizing microstructures that programmatically modify and extend cell surface architecture” and “cell-anchored filaments are sensitive flow rate meters” would in my opinion strengthen the conclusions of the paper.

We really appreciate these suggestions to strengthen the paper. We have modified the wording of this section and used the type of specific wording the reviewer suggests.

6. To the point above, there are reported examples of functionalizing living cell membranes, such as the work by Akbari et al, Adv Mater, 2017 where synthetic DNA scaffolds were anchored to the surfaces of adherent, suspension, and primary cell types on which numerous functions such as directed cell-cell contacts and fluorescence readouts could be programmed.

We agree that this paper is relevant to this work and have added a reference to it in the section of the introduction mentioned above in point 5.

Comments on the Results (reliably anchoring nanotube seeds to cell receptors)

7. In the results section, the authors mention that the number of attached seeds per cell was 107 ± 17 seeds for 64 pM seeds. However, it is not very clear how they came to that estimation. Looking at the supplementary material, it looks like the authors were able to measure the average fluorophore intensity per cell. However, making a direct link between the intensity and the

number of seeds can be misleading. A better estimate can be the average concentration of seeds per cells, since this is the standard for the titrations made.

We appreciate the reviewer bringing this up and carefully considering how measurements might be made in this case. We agree that attention should be paid when creating a corresponding number of seeds and a fluorescence intensity. In the case of the nanotube seeds, creating a correspondence was possible because each seed is extremely bright, as it is labelled by 100 small-molecule fluorophores, and is also almost >50 nm in length. As such, individual seeds can be readily observed on the cell surface. That said, such a correspondence is a conservative estimate given that seeds that are overlapping cannot always be counted as two separate objects. In the revised version of the manuscript we have updated the main text to make this limitation in the correspondence clear. We have also added a significant amount of detail about how we developed this correspondence, in particular by described our method for counting the number of seeds number per cell in Supplementary Note S6.2. the result of our analysis were also added to Supplementary Figure S4. Supp Figure S4 makes clear that the number of measured seeds and the fluorescence intensity values per cell were strongly correlated in our different experiments, validating the measurement approach we used.

8. In the EGFR-EGFR antibody binding scheme, is there any reason why the authors did not increase the concentration of the seeds to nM scale to show that the limiting factor was indeed the affinity?

The reviewer brings up a great point here – showing that when seeds were provided at a concentration above the K_D value of the antibody would provide strong support for the idea that the seeds' attachment was affinity limited when using antibodies to direct attachment. In practice, however, performing a conclusive experiment to test this claim would be difficult. Based on the concentrations of reagents that we have (which are procured from standard vendors), it would be difficult to introduce seeds at concentrations >10 nM and would require a totally different approach above about 25 nM. The reference we cite¹ lists the K_D value of the antibody as between 2 and 15 nM, and states that it is not easily correlated with cell type or total number (concentration) of EGF receptors. We thus did not feel such an experiment would be conclusive if we did not see significant binding.

More generally, while seeds might be provided at nanomolar concentrations, no one to our knowledge has prepared seeded nanotubes at a concentration of above 1 nM of seeds. Generally, protocols for growth of nanotubes from seeds use concentrations of seeds in the range of 1-100 pM². These concentrations reflect the means by which seeded nanotubes are assembled (seeds serve as nucleation templates and the monomers must both be in great excess to the seeds to produce micron-length nanotubes) as well as at a concentration low enough to allow reversible assembly of monomers as well as their very large size (Supp Figure S5b); nanotubes are microns in length and thus at least an order of magnitude larger than the DNA seeds which themselves are large complexes. A key goal in this work was to develop strategies for conjugating seeded nanotubes to receptors given these limitations, and, given that the preparation of other complex devices would also be subject to similar limitations. Receptors with low copy numbers would present similar challenges as well.

To help clarify the concentrations used and address these points, we now list the concentrations of antibodies and other proteins used in our experiments in both mg/ml and nM in Supplementary Note S18.

9. Is there a reason for the 16 and 64 pM seed concentration for the AMDA scheme?

We appreciate the reviewer bringing this up. The reasoning is similar to the reasoning described in this referee's point 8 above, reflecting the seeded nanotubes' large size and standard methods of preparation. We chose 16 and 64 pM among the concentrations we used in tests of other attachment schemes to provide a consistent basis of comparison with earlier experiments. We chose 64 pM with the idea that the highest concentration would provide the best chance of success and 16 pM to provide a second value to test the effects of concentration that provided a clear contrast to 64 pM (but still a feasible chance of success). Our kinetic analysis in Supp. Note S12 (and Supp. Note S17) also suggested that these concentrations could be used successfully for DNA's on rate of hybridization. We have now added a sentence to the text about our reasoning.

10. Is there a reason why the quantification of the nanotube attachment to HEK293 suspended cells was done using flow cytometry as opposed to direct quantification from confocal images? How do the confocal results compare to the flow cytometry?

We appreciate the reviewer for bringing this point up here and give us the opportunity to explain our different quantification methods. Flow cytometry is a widely used method to analyze and characterize cells by measuring their fluorescence. Considering that the HEK293 cells were in suspension, the use of confocal microscopy would have been lower throughput and more laborious than for adherent cells. Further, flow cytometry allowed us to collect much more data: a single flow cytometry experiment allowed us to measure the seeds fluorescent intensity on around 10000 cells in a single measurement. We did not use flow cytometry to measure the fluorescence intensity of the nanotubes, because despite these advantages the shear stress on cells during the process during flow cytometry could break the micron-length nanotubes.

For seeds, however, we found that the two techniques provided comparable results. Supp. Note S24.1 describes a new experiment we added to the revised version of the manuscript in which we measure the fluorescence intensity of seeds on suspended HEK293 cells after AMDA and after a control process. These results confirm those determined by flow cytometry.

11. On a similar note, what is the reason of using two different standards for reporting the anchoring of DNA nanotubes in figure 3 c, f (i.e. mean fluorescence intensity vs. seed count)

Interesting question, and we appreciate the very thorough reading here! Our approach in 3c and f followed the goal of being consistent in our measurement approach if possible, but taking an alternative approach if it provides more or clearer information. Fig 3c used the approach we used for seeds, and allowed us to find that seed attachment remained reliable even when nanotubes were attached to seeds (in comparison with Fig 2c). When looking at images of the HEK293 cells, which were suspended, it was apparent that each nanotube could be seen individually. This is a more direct measure than in 3c so we opted for this approach. Given that

counting the numbers of seeds and the fluorescence intensity did not change the outcome (Supp Figure S4), and especially, the very large difference in numbers between the control and the AMDA samples, we do not expect that this approach has an impact on the interpretation of the data in the narrative.

We also quantified the average fluorescence intensity of seeds/nanotubes per HEK293 cell and found that the two methods provided similar results (Supp. Figure S14). We have added a new experiment to the revised version of the manuscript, described in Supp. Note S29.1, in which we measure the fluorescence intensity of the seeds/nanotube on suspended HEK293 cells after AMDA and after a control experiment.

12. In measuring the stability of DNA nanotubes on cell surface, what is the effect of the cell suspension buffers on the stability of the nanotubes?

The reviewer brings up a great point here. In this experiment, the cell suspension buffer was the cell growth medium which was the DMEM medium containing 10% FBS and 1% penicillin-streptomycin and we measured the lifetimes of seeded nanotubes on the cell membrane for 6 hours. Our lab has previously studied the stability of seeded nanotubes anchored on the glass surface in the cell growth medium and found that the lengths of the seeded nanotube were not changed obviously during first 6 hours³, a result consistent with other work on the stability of large DNA nanostructures under different conditions relevant for biological studies and bioengineering. Based on these results, it seems likely that the cell growth medium did not have a major effect on the stability of nanotubes during the experiment that we measured lifetime of nanotubes on HeLa cell. In the revised version of the paper we now mention this reasoning behind our choice of buffers in the description of the protocol for this experiment (Supplementary Note S32) and discuss the idea that the problem of serum degradation of nanotubes is relevant for longer timescales in the conclusion of the paper.

13. On a similar note, what is the effect of the anchoring of DNA nanotubes, and the later addition of methylcellulose, on cell mortality?

We appreciate the reviewer bringing this up. For devices to be useful in the context of living cells, they must have a minimum of deleterious effects. In our observations of cells which have DNA nanotubes anchored on them, we have not observed either any increases in cell mortality or other harmful effects. All the reagents used in the AMDA process, including DNA, PEG20K⁴, antibodies, neutravidin and streptavidin⁵ are all biocompatible materials and were used at concentrations and over incubation times that are not toxic to mammalian cells. In fact, one advantage of our approach is the ability to use very low concentrations of each of the reagents, as discussed both in the paper and in response to your question 8 above. In the revised version of the paper we now mention the point that the materials used to construct the device are biocompatible and were used at low concentrations.

During experiments in which we measured the nanotube lifetime on HeLa cell membrane, we continuously monitored cells and did not observe any obvious decrease of cell viability. An experiment in which cells were cultured overnight after nanotube attachment (Supp. Note S32 and Supp. Figure S16) also did not turn up any deleterious effects.

Similarly, methylcellulose is a non-toxic and biocompatible polymer which has been widely used in cell culture, stem cell differentiation⁶ and tissue engineering^{7,8}. For example, 1% methylcellulose medium has been used for colony formation assays of hematopoietic stem/progenitor cell and 3% methylcellulose medium is used to trigger the rapid formation of multicellular spheroids⁹. During these uses, no effects of methylcellulose on cell mortality were reported. Here we used 0.6% methylcellulose medium and, consistent with prior reports, did also not observe the addition of methylcellulose affect the cell mortality.

14. “>40±4.8% of the resulting filaments were >3 μm long.” An explanation of the significance of this exact length would be fitting.

We appreciate the reviewer bringing this up. We chose 3 μm as a threshold length for reporting to emphasize that the nanotubes were long enough that their angles and contours could be readily visualized as this length is far greater than the diffraction limit. This length further underscores that the characterization of these devices is apt. We have now altered the main text to include this point. As a 3 μm is also a bit longer than the average length of a primary cilium¹⁰.

15. Supp. Figure S5b: Is there a reason incubation couldn't be extended to 3 days? It appears that approximately 40% of filaments grown in these conditions were also more than 3 μm but was left out of the main text.

This is a good question. Supp. Figure 5b gives the length distribution of the seeded nanotubes without PEG coating after 1 day of incubation at 37°C. In previous studies, we found that 3 days of incubation did not increase their lengths over this process.³ As a result, we usually only incubated seeded nanotubes without attached PEG for 1 day. In Supp. Figure S5, our intent was to show that PEG coated seeded nanotubes had a similar distribution of lengths as normal seeded nanotubes do after a full incubation, not a comparison of the effects of incubation time; we apologize for any confusion. This should also explain why we did not further discuss the length distribution of seeded nanotubes without PEG coating in the main text, as these materials were not used in the cell attachment experiments or any other experiments the main text provides. We have attempted to clarify this point further in the caption to Supp. Figure S5b.

(Comments on the Results (shear stress study):

16. What is the height of the flow chamber? In the main text, it was mentioned to be 400 μm height. However, in the caption of figure 4, it was mentioned to be 0.54 mm in height.

We thank the reviewer for pointing out this confusion and giving us the opportunity to correct it here. For the height of the flow chamber, we actually used flow chambers with different heights for 1) measuring the bending of the seeded nanotubes anchored on substrate surface and 2) for measuring the bending of the seeded nanotubes anchored on cell membranes. For the experiment in which nanotubes were anchored to a glass substrate, we used glass bottom flow chamber from Ibidi height 0.54 mm, length 17 mm and width 3.8 mm. For experiments characterizing the bending of nanotubes anchored on cell membranes, we used a plastic-bottom chamber also from Ibidi with height 0.4 mm, length 17 mm and width 3.8 mm. The reason for this difference was historic – we performed the cell-anchored nanotube experiments first using a plastic-bottomed dish.

When we sought to perform the experiments to measure bending of nanotubes on a surface as a control, we found that we could not easily functionalize the plastic surface to attach nanotubes. We therefore used a glass flow cell with the closest dimensions to the plastic cell used first that Ibidi offered. The differences in these chambers are minor, but we agree that not explicitly pointing this out caused confusion. We have now updated the text to provide this information in the Methods once and to clearly describe this distinction.

We have repeated the simulations using the height 0.54mm and updated all of the relevant figures and descriptions in Supplementary Note S34 to describe this single height.

17. On a similar note what are the dimensions for the flow chambers used for the cell experiments? What is the reason for the difference between the formulas 4.31 and 4.32 in the supplementary material?

We appreciate the reviewer bringing this up. As we mentioned above, the heights of the flow chambers used for the two types of experiments in which we characterized nanotube bending angles were slightly different. In the Supplementary Note S35, the formulas 4.31 and 4.32 are the two formulae provided by the manufacturer to calculate the shear stress as a function of flow rate through the two respective channels.

$$\tau = \eta \times 104.7 \times Q \quad (4.31)$$

$$\tau = \eta \times 176.1 \times Q \quad (4.32)$$

where τ = shear stress (dyn/cm²), η = dynamical viscosity (dyn s/cm²), and Q = flow rate (mL/min). To clarify this point we now explicitly give the dimensions of each of the types of flow cells in the Supplementary Information and give the reason for the difference in the formulas.

18. Minor comment: was the flow channel surface's hydrophobicity/hydrophobicity taken in consideration while assuming a uniform flow?

We thank the reviewer for this question. Here don't take the channel surface hydrophobicity in consideration while assuming a uniform flow. In the model, we assume uniform flow in the width and length direction and a non-slip boundary condition in the height direction. The height of the channels are each on the order of 500 microns, which is large enough that hydrophobicity/hydrophobicity does not affect the flow profile.

19. On a similar note, for the cell surface measurements, were the disruptions in the flow profile due to the presence of cells taken in consideration while estimating the shear stress? It appears like the shear stress reported in this case was still the wall shear stress as opposed to the estimated shear stress on the cell surface.

This is a good question! For the cell surface measurements, the disruptions on the flow profile due to the presence of cell were not taken into consideration while estimating the shear stress. This is because the heights of the adherent HeLa cell are on average 7-10 μm , very small

compared with the height of the flow chamber (400 μm). In the Supplementary Note S34.1, for a Stock flow, the velocity profile is formulas 4.4:

$$U(z) = \frac{6Q}{H^3W}(Hz - z^2)$$

H and W is the chamber height and width, Q is the volumetric flow rate.

So the shear stress is:

$$\tau = \mu \frac{\partial U(z)}{\partial z} = \mu \frac{6Q}{H^3W}(H - 2z)$$

μ is the dynamic viscosity of the fluid.

From this equation, one can see that the shear stresses along the cell surface, *i.e.* the shear stresses for z from 1-10 μm are quite close to those at z=0, which is the wall shear stress. It is therefore reasonable to approximate the shear stress at the cell surface as the same as the shear stress at the fluid cell wall. We have now noted this point in Supplementary Information S43.1.

20. How were the nanotubes on top of the cells selected? Since the HeLa cells are of irregular topography, what defines the top of the cell?

This is a good question. We defined that the top of the cell as the part of the cell farthest from substrate (*i.e.* glass) under the spinning confocal microscope in the cell labeling channel, as defined by the Z height at which the part of the cell appeared in confocal images. The seeded nanotubes anchored on this part of each cell were selected and imaged. We have now noted this point in Supplementary Information S40.

21. Looking at figure S22, it seems like the angles Φ_{upper} and Φ_{lower} were measured from different points on the y-axis. Assuming that the seed doesn't move over time, would not it be more accurate to measure both angles with respect to the same anchoring point? (*i.e.* the seed)

We appreciate the reviewer bringing this up and give us the opportunity to explain it here. We actually measured the two angles from the same anchoring point – we simply offset the diagrams showing how the angle is defined from the anchoring point in each of the subfigures. The anchoring point was the intersection of the two straight lines along the edges of the sector area formed by the nanotube time projection images. We have now also labeled the anchoring points in the example image in Supp. Figure S25.

22. What was the criteria for selecting the nanotubes to be analyzed at each shear stress value? Did they have to have a similar X-Y projection length? What happens with the nanotubes that appear as dots at lower flow rates?

We appreciate the reviewer bringing this up and giving us the opportunity to clarify it here. The main criteria used for selecting a nanotube to be analyzed were that a nanotube be a) longer than 2 μm and b) not evidently sticking to a surface or cell membrane. Nanotubes that overlapped with others or whose anchor location moved during imaging were also excluded. We selected nanotubes longer than 2 μm because short nanotubes appeared as dot in X-Y projection images,

so it was hard to measure the total rotation angle. We have now added information about these selections to SI Note S41 and now also mention in the main text the importance of using long nanotubes for more precise measurements of shear stress.

As we mentioned above, we excluded seeded nanotubes that were too short and appeared as dots in the X-Y projection images under fluid flow. In the range of shear stresses we considered, even under the lowest fluid shear stress 0.05 dyn/cm^2 the seeded nanotubes all bended in the flow direction and remain in X-Y plane. We therefore could measure the nanotubes' movement using the methods and criteria described as above.

Comments on the Results (Growing nanotubes on living cells)

23. Reasoning for using 4PEG nanotubes stated in Supp. Note 44 should also be stated in main text. They provide “higher attachment and detachment rate than the 6nt seeded nanotubes monomers.”

We appreciate the reviewer bringing this up; increasing the rates of kinetic exchange was the reasoning behind this change. That said, DNA hybridization kinetics suggests that shortening the base pair length of the sticky ends should increase the detachment rate of nanotube monomers, but not the attachment rate. (These observations are also consistent with measurements of nanotube monomer on and off rates.¹¹)

We now specifically note that we made the change in sticky end length to increase the monomer detachment rate in the main text.

24. A histogram of average lengths of nanotubes after extension would be a good visual to include.

We appreciate the reviewer bringing this up and we have added figure panels depicting the length distributions of the seeded nanotubes anchored on the glass surface and cell membrane after end-to-end nanotube joining to Supplementary Figure S31 and add the according methods for collecting the data for these histograms to the Supplementary Note S45 and S46.

25. Are there applications for the random specificity of end-to-end joining of nanotubes (Figure 5b)?

This is an interesting question. The idea behind the design of the experiment in which we anchored nanotubes with one type of fluorophore on cells and added nanotubes and monomers labeled with a different dye was intended to produce a visually distinctive pattern on the nanotubes if joining has occurred rather than itself to be a useful method. The joining process, generally, is an important demonstration that devices can be dynamic – can change form over time or heal, and the dynamic growth and recovery of biological filament structures suggestions many potential applications for these ideas.

That said, one might also envision applications specifically for the patchy structures that result from the joining process, such as in speckle microscopy that has been used to track and characterize filaments.¹² Heterogeneous structures might combine multiple chemically functional groups on devices. We've now mentioned this point in the conclusions.

26. Maybe this isn't possible in the presence of certain flow rates, but since growing nanotubes can extend to micron-lengths, have any nanotube loops been observed e.g. shown below?

This is a good question. We did not see any nanotube loops either after initial periods of incubation or after nanotube end-to-end joining under our observation. One important reason for this is that anchored nanotubes only have one free end and therefore cannot join. The nanotubes that were being joined in some cases had two free ends, so could join, and such loops have been observed previously.¹³ However, given that the nanotubes have a persistence length of $8.7 \pm 0.5 \mu\text{m}$, such loops generally form from nanotubes much longer than the generally 2-3 μm long nanotubes used in this study. We did, however, see side-to-side joining of nanotubes, induced by the crowding agents used and have tried to make this point more salient.

Comments on the conclusions:

27. Data interpretation is somewhat limited. Discussion could include a summary of the main results, limitations of a low fraction of seeded nanotubes remaining on the cell surface after several hours and not being able to control the final length of growing nanotubes, and how AMDA compares to and leads other current antibody mediated DNA attachment devices, if any.

We appreciate the reviewer bringing this up and we have made significantly changes to the Conclusion section to try to address these important questions.

Comments on the figures and supplementary movies:

28. Figure 1: (A) The cell receptors are hard to see against the background. Changing the color might help making them more distinguishable. (B) The panel would benefit from additional annotation to show that radial and longitudinal polymerization of the nanotubes. (C) What are the dimensions for the seeds?

We appreciate the reviewer bringing this up and we made these changes to Figure 1. The color of the receptor has been changed, we added a panel to b and d to show the lattice structure and therefore how polymerization could occur, and also added the lengths of the seed (the widths of the seeds and the nanotubes are similar).

29. Figure 2: Are these results for the 64 pM seed concentration? If yes, please add that to the caption

We appreciate the reviewer bringing up this important detail. We have now added the concentrations of seeds used in each of the processes whose results are shown in Figure 2; references to full methods are also given for each experiment which include the concentration of each reagent, complete protocol details such as incubation times.

30. Figure 4: (A) The nanotube deflection is barely visible. (D, E) The confocal images in panels d and e should have axes

We appreciate the reviewer bringing these points up, as they will definitely improve the paper. We have made the nanotube deflection more salient with an inset and added axes to d and e.

31. Supplementary Figure S5: Looking at the sample epi-fluorescence of the seeds after incubation, is there a reason why the polymers look more coiled after 3 days of incubation as compared to just one day even though there is not a big shift in the length distribution? Were the same samples imaged after 1 and 3 days of incubation?

We appreciate the reviewer bringing this up and giving us the opportunity to clarify it here. In Supplementary Figure S5 the samples shown in the two panels actually are different types of samples. One depicts seeded nanotubes assembled from monomers without PEG conjugates imaged after 1 day of incubation while the other depicts seeded nanotubes with PEG conjugates imaged after 3 days of incubation. We definitely agree with the reviewer the PEG-coated seeded nanotubes appear a bit coiled after they are deposited on the coverslips for imaging, and this appearance does not appear dependent on the time of incubation. We updated the caption of this figures be clearer about this distinction.

32. Time stamps should be added to Supp. Movies 3 and 4

We appreciate the reviewer bringing this up. We have now added time stamps to Supp. Movie S3. But we could not add time stamps for Supp. Movie S4 since we were not sure the time intervals between two images. This movie was generated by taking 20 continuous cycles in which fluorescence images in the Cy3, atto647 channels and a bright field image were captured without latency. We usually use this method to quickly get a movie to record the nanotube movements.

References

- 1 Zhou, Y. *et al.* Impact of Intrinsic Affinity on Functional Binding and Biological Activity of EGFR Antibodies. *Molecular Cancer Therapeutics* **11**, 1467-1476, doi:10.1158/1535-7163.Mct-11-1038 (2012).
- 2 Schaffter, S. W., Scalise, D., Murphy, T. M., Patel, A. & Schulman, R. Feedback regulation of crystal growth by buffering monomer concentration. *Nature Communications* **11**, 6057, doi:10.1038/s41467-020-19882-8 (2020).
- 3 Li, Y. & Schulman, R. DNA Nanostructures that Self-Heal in Serum. *Nano Letters* **19**, 3751-3760, doi:10.1021/acs.nanolett.9b00888 (2019).
- 4 Liu, G. *et al.* Cytotoxicity study of polyethylene glycol derivatives. *RSC Advances* **7**, 18252-18259, doi:10.1039/C7RA00861A (2017).
- 5 Jain, A. & Cheng, K. The principles and applications of avidin-based nanoparticles in drug delivery and diagnosis. *Journal of Controlled Release* **245**, 27-40, doi:<https://doi.org/10.1016/j.jconrel.2016.11.016> (2017).
- 6 Kaufman, D. S., Hanson, E. T., Lewis, R. L., Auerbach, R. & Thomson, J. A. Hematopoietic colony-forming cells derived from human embryonic stem cells. *Proceedings of the National Academy of Sciences* **98**, 10716-10721, doi:10.1073/pnas.191362598 (2001).
- 7 Thirumala, S., Gimble, J. M. & Devireddy, R. V. Methylcellulose Based Thermally Reversible Hydrogel System for Tissue Engineering Applications. *Cells* **2**, 460-475 (2013).

- 8 Ahlfeld, T. *et al.* Methylcellulose – a versatile printing material that enables biofabrication of tissue equivalents with high shape fidelity. *Biomaterials Science* **8**, 2102-2110, doi:10.1039/D0BM00027B (2020).
- 9 Kojima, N., Tao, F., Mihara, H. & Aoki, S. in *Stem Cells and Cancer in Hepatology* (ed Yun-Wen Zheng) 145-158 (Academic Press, 2018).
- 10 Dummer, A., Poelma, C., DeRuiter, M. C., Goumans, M.-J. T. H. & Hierck, B. P. Measuring the primary cilium length: improved method for unbiased high-throughput analysis. *Cilia* **5**, 7, doi:10.1186/s13630-016-0028-2 (2016).
- 11 Hariadi, R. F., Yurke, B. & Winfree, E. Thermodynamics and kinetics of DNA nanotube polymerization from single-filament measurements. *Chem Sci* **6**, 2252-2267, doi:10.1039/c3sc53331j (2015).
- 12 Waterman-Storer, C. M., Desai, A., Chloe Bulinski, J. & Salmon, E. D. Fluorescent speckle microscopy, a method to visualize the dynamics of protein assemblies in living cells. *Current Biology* **8**, 1227-S1221, doi:[https://doi.org/10.1016/S0960-9822\(07\)00515-5](https://doi.org/10.1016/S0960-9822(07)00515-5) (1998).
- 13 Axel Ekani-Nkodo, A. K., Deborah Kuchnir Fygenon. Joining and Scission in the Self-Assembly of Nanotubes from DNA Tiles. *Physical Review Letters* **93**, 268301, doi:10.1103/PhysRevLett.93.268301 (2004).

Reviewers' Comments:

Reviewer #1:

Remarks to the Author:

My concerns have been addressed. This is an excellent manuscript and I recommend acceptance as is.

Reviewer #3:

Remarks to the Author:

The authors have done a commendable job in the revision. I do not have any further comments on the manuscript and my recommendation would be to accept it.